# CTP sensing and Mec1$^{ATR}$-Rad53$^{CHK1/CHK2}$ mediate a two-layered response to inhibition of glutamine metabolism

Arta Ajazi[1]*, Ramveer Choudhary[1], Laura Tronci[1,2], Angela Bachi[1], Christopher Bruhn[1]*

**1** The FIRC Institute of Molecular Oncology (IFOM), Milan, Italy, **2** IRCCS San Raffaele Scientific Institute, Molecular Basis of Cystic Kidney Diseases, Division of Genetics and Cell Biology, Milan, Italy

* arta.ajazi@ifom.eu (AA); christopher.bruhn@ifom.eu (CB)

## Abstract

Glutamine analogs are potent suppressors of general glutamine metabolism with anti-cancer activity. 6-diazo-5-oxo-L-norleucine (DON) is an orally available glutamine analog which has been recently improved by structural modification for cancer treatment. Here, we explored the chemogenomic landscape of DON sensitivity using budding yeast as model organism. We identify evolutionarily conserved proteins that mediate cell resistance to glutamine analogs, namely Ura8$^{CTPS1/2}$, Hpt1$^{HPRT1}$, Mec1$^{ATR}$, Rad53$^{CHK1/CHK2}$ and Rtg1. We describe a function of Ura8 as inducible CTP synthase responding to inhibition of glutamine metabolism and propose a model for its regulation by CTP levels and Nrd1-dependent transcription termination at a cryptic unstable transcript. Disruption of the inducible CTP synthase under DON exposure hyper-activates the Mec1-Rad53 DNA damage response (DDR) pathway, which prevents chromosome breakage. Simultaneous inhibition of CTP synthase and Mec1 kinase synergistically sensitizes cells to DON, whereas CTP synthase over-expression hampers DDR mutant sensitivity. Using genome-wide suppressor screening, we identify factors promoting DON-induced CTP depletion (TORC1, glutamine transporter) and DNA breakage in DDR mutants. Together, our results identify CTP regulation and the Mec1-Rad53 DDR axis as key glutamine analog response pathways, and provide a rationale for the combined targeting of glutamine and CTP metabolism in DDR-deficient cancers.

## Author summary

Cancer cell proliferation is supported by high metabolic activity. Targeting metabolic pathways is therefore a strategy to suppress cancer cell growth and survival. Glutamine is a key metabolite that supports a plethora of anabolic, growth-promoting reactions in the cell. Therefore, the use of small molecules that block glutamine-dependent reactions has been extensively investigated in cancer therapy. Knowledge about the pathways that influence sensitivity towards glutamine metabolism inhibitors would help to tailor the use of such glutamine-targeting therapies. In this study, we use budding yeast as model system

**Data Availability Statement:** All raw data files are available from the Mendeley Data database: https://data.mendeley.com/datasets/9rkz7k8ktj/1.

**Funding:** This work was supported by the Associazione Italiana per la Ricerca sul Cancro (AIRC, https://www.airc.it/). A.A. was supported by fellowship ID14974, and C.B. was supported by fellowship ID16173. Funding for the open access charge was provided by AIRC (ID16173). The funders had no role in study design, data collection and analysis.

**Competing interests:** The authors have declared that no competing interests exist.

to identify the pathways that mediate or restrict the toxicity of a representative inhibitor of glutamine metabolism, the glutamine analog 6-diazo-5-oxo-L-norleucine (DON). We describe a response mechanism mediated by an inducible CTP synthase that promotes nucleotide homeostasis during DON exposure to prevent DNA breaks. Moreover, we show that combined inhibition of the inducible CTP synthase and DNA damage response enhances DON toxicity, pointing out a potential therapeutic application in cancers with defective DNA damage response.

## Introduction

Glutamine is utilized in many metabolic processes such as the biosynthesis of pyrimidine and purine nucleotides, glutathione and non-essential amino acids, and the replenishment of tricarboxylic acid (TCA) cycle metabolites. Cancers commonly depend on glutamine supply as nitrogen and carbon source for rapid proliferation [1]. Hence, various steps in glutamine metabolism have been explored as therapeutic targets to reduce intracellular glutamine concentration or inhibit glutamine-dependent reactions [2].

Glutamine mimetics inhibit glutamine-dependent reactions and efficiently kills tumor cells *in vivo* [3], but their use is hampered by considerable toxicity [4]. While prodrug strategies [4–6] and combination treatments [7,8] are currently explored to reduce their side effects, knowledge on the major sensitivity and resistance mechanisms may improve drug combination designs and treatment rationale. However, the genetic determinants of glutamine mimetic sensitivity and resistance have not been systematically mapped.

6-diazo-5-oxo-L-norleucine (DON) is a glutamine analog with anti-tumor activity [3]. Its recently developed prodrug JHU-083, a DON precursor which is converted to the active compound at the tumor site, facilitates the establishment of a therapeutic window in mouse tumor models to target cancers [5,6], and may thus be evaluated as glutamine analog for cancer treatment. DON inhibits at least 10 glutamine-utilizing metabolic enzymes operating in the synthesis of nucleotides, amino acids, hexosamines and $NAD^+$, as well as the conversion of glutamine to glutamate (glutaminolysis) to fuel biosynthetic reactions through mitochondrial metabolites of the tricarboxylic acid (TCA) cycle [4,9]. Thereby, DON globally blunts metabolic reactions required for rapid division and inhibits growth-promoting mechanistic target of rapamycin complex 1 (mTORC1) signaling, which is stimulated in glutamine-utilizing cancer cells by glutaminolysis [10]. The general inhibition of glutamine metabolism by DON is thought to narrow down the window of potential resistance mechanisms [6], which are documented for specific glutaminolysis inhibition and potentially involve over-expression of the glutamine synthetase GLS2 and metabolic reprogramming to utilize tricarboxylic acid (TCA) cycle inputs different from glutamine [11–13].

Inhibition of nucleotide synthesis is a common strategy in cancer treatment to target DNA synthesis, which preferentially kills rapidly replicating cells [14]. Nucleotide shortage causes critically low deoxy-nucleotide pools which impair the progression of DNA polymerases in S phase [15]. To avoid replication-associated DNA damage, an evolutionarily conserved replication stress response pathway detects DNA structures at aberrant replication forks and triggers a signaling cascade to stabilize forks [16], enhance deoxy-nucleotide synthesis [17,18] and halt cell cycle progression [19,20]. The DNA damage response (DDR) kinases ATR (yeast: Mec1) and CHK1 (yeast: Rad53) are the key mediators of this pathway [21]. While complete gene deletions cause extensive chromosome breaks [22] and result in early embryonic lethality

[23,24], inhibition of their kinase activity hyper-sensitizes towards replication stress and has been widely explored for cancer therapy [25].

Here we delineate the genetic network of key glutamine analog resistance and sensitivity genes using chemogenomic screening in budding yeast. We show that five evolutionarily conserved proteins operating in pyrimidine synthesis (Ura8[CTPS1/2]), purine salvage (Hpt1[HPRT1]), regulation of glutamine metabolism (Rtg1) and replication stress response (Mec1[ATR] and Rad53[CHK1/CHK2]) are crucial for resistance to the glutamine analog DON. We further find that specific amino acid transporters and TORC1 activity are required for full DON toxicity, and their inhibition ameliorates DON hypersensitivity of DDR mutants. We demonstrate that combined targeting of CTP synthase and replication stress response kinases synergistically sensitizes cells towards DON. We further unravel an adaptive glutamine analog response mediated by metabolite sensing at the *URA8*[CTPS1/2] gene. Our results provide approaches for the enhancement of glutamine analog efficiency with potential implications in DDR-deficient cancers.

## Results

### Identification of glutamine analog resistance genes

To identify genetic factors of glutamine analog resistance, we screened the Yeast Knockout (YKO) budding yeast library with deletion of non-essential genes [26] for mutants with altered sensitivity towards the glutamine mimetic 6-diazo-5-oxo-L-norleucine (DON) (Fig 1A and S1 Table). We validated the candidates in independent clones derived from the crossing of library mutants with a wildtype (wt) strain (S1A and S1B Fig, S2 Table). In total, we identified three strongly sensitive (*ura8Δ*, *rtg1Δ*, *hpt1Δ*) and two mildly sensitive (*rhb1Δ*, *par32Δ*) mutants (Figs 1A and 1B and S1B), implying functions of the deleted genes in mediating glutamine analog resistance. The only mutant classified as resistant (*fui1Δ*) was mildly slow-growing on normal media, and its growth on DON was indistinguishable from the wt control (Fig 1B). Hence, while the relative growth-inhibiting effect of DON is weaker in *fui1Δ* mutants, the *fui1Δ* mutation does not confer an absolute growth advantage on DON. The genes deleted in the strongly sensitive strains were linked to glutamine-related processes: *URA8* encodes one of two CTP synthases (human: *CTPS1*, *CTPS2*), which catalyze the glutamine-dependent conversion of UTP to CTP and are directly and irreversibly inhibited by DON [27]; *HPT1* (human: *HPRT1*), encodes the hypoxanthine-guanine phosphoribosyltransferase which facilitates purine salvage, a parallel pathway to the DON-sensitive purine *de novo* synthesis [28]; *RTG1* encodes a TOR-inhibited transcription factor which is activated by glutamine starvation and promotes the expression of glutamine synthesis genes [29]. The weaker chemogenomic DON interactors, Rhb1 (a putative Rheb-like GTPase involved in trans-membrane transport), Par32 (an adapter protein that links TORC1 activity to amino acid transporter localization), and Fui1 (a uridine permease), control uptake processes and may affect DON or glutamine uptake. In summary, CTP synthase, purine salvage and the Rtg1 transcriptional program are the major non-essential mediators of glutamine analog resistance in budding yeast.

### The minor CTP synthase Ura8 mediates a glutamine analog response

Glutamine is required for the *de novo* synthesis of nucleotides and hence deoxynucleotide triphosphates (dNTPs). The identified DON resistance genes may hence mediate resistance to the shortage of glutamine, nucleotides or dNTPs. To stratify the roles of the candidates, we treated the mutant cells with hydroxyurea (HU), an inhibitor of the ribonucleotide reductase, which causes a nearly complete depletion of the purine deoxynucleotide dATP and a partial depletion of the other dNTPs [30]. Among the three strongest candidates (*rtg1Δ*, *hpt1Δ*,

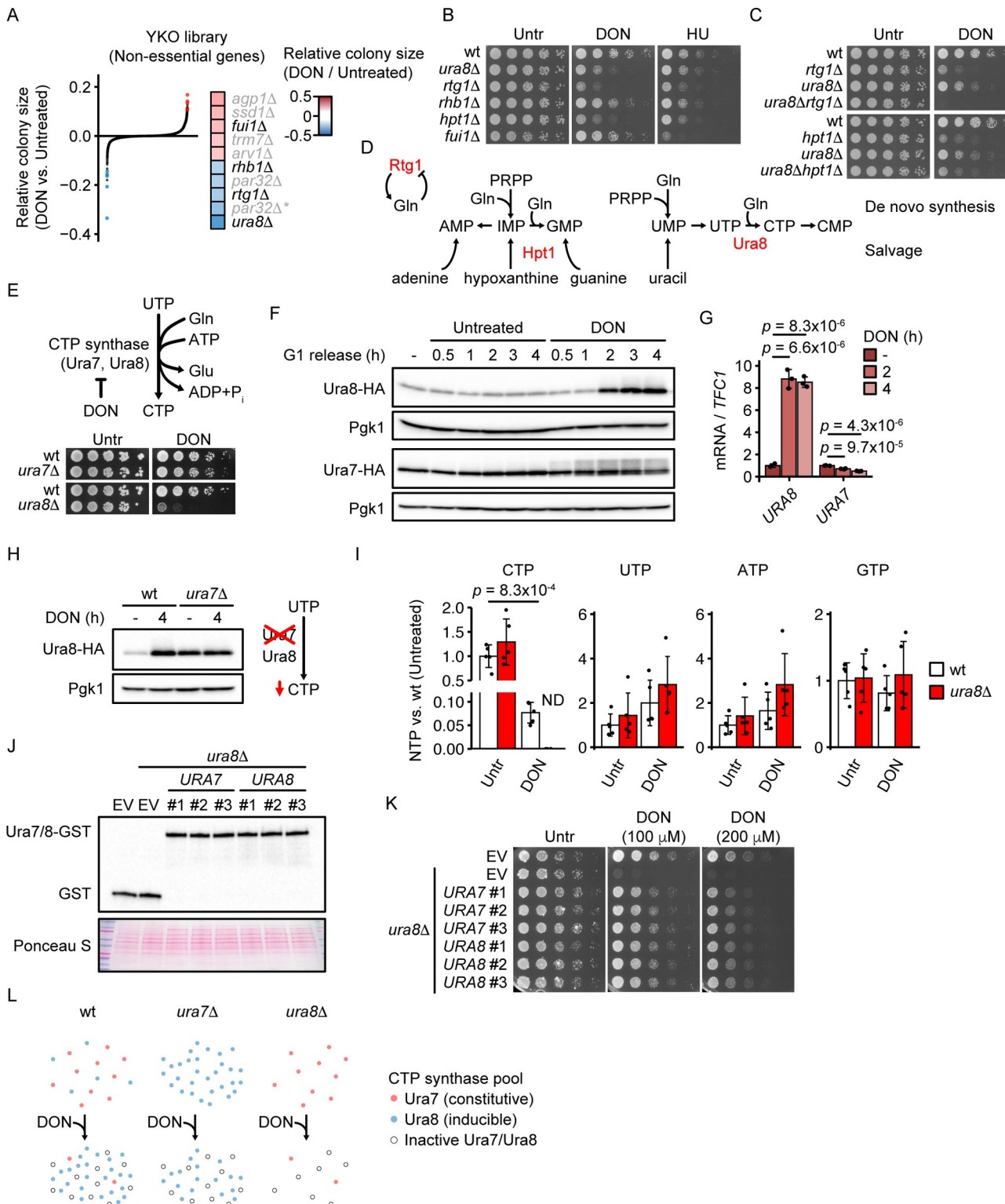

**Fig 1. Identification of glutamine analog resistance pathways.** (A) Screening for genes involved in resistance to the glutamine analog DON. The viable haploid gene deletion yeast library was screened for DON-sensitive deletion mutants by replicating library strains on YPD (Untreated) or YPD + 200 μM DON. Each dot represents a library mutant. The y axis represents the difference between DON-treated vs. untreated colony size. The x axis represents the gene rank from sensitive (left) to resistant (right). All values are $\log_2$-transformed, corrected for standard deviation and normalized to the 70-percentile of all library mutants. Significant sensitivity or resistance to DON was determined by separation of DON-treated and untreated colony

mean sizes by at least two standard deviations and is indicated in blue (sensitive) and red (resistant). The candidate identities are shown in the right panel. Validated and not confirmed candidates are written in black and grey letters, respectively. Statistical information is provided in S1 Table. (B, C) $10^7$ cells / mL of the indicated genotype in the S288C background were serially diluted (1:6), spotted on YPD plates with or without 250 µM DON or 200 mM HU and grown for 2d (Untreated), 3d (DON) or 4d (HU). (D) Pathway scheme illustrating the relation of the major DON resistance genes (red). (E) $10^7$ cells / mL of the indicated genotype in the W303 background were serially diluted (1:6), spotted on YPD plates with or without 150 µM DON and grown for 2d. The scheme illustrates the reaction catalyzed by CTP synthases Ura7 and Ura8. (F) Cells expressing endogenous HA-tagged Ura8 or Ura7 were arrested in G1 phase with α-factor and released into YPD with or without 300 µM DON. Cells were fixed in TCA at the indicated time points and proteins were analyzed by Western blot. (G) Wildtype cells grown in YPD medium were treated with 300 µM DON for the indicated duration. Samples were collected to quantify *URA8* and *URA7* mRNAs by RT-qPCR using *TFC1* as reference (n = 3 independent replicate cultures). Significances were calculated with 1-way ANOVA ($p_{URA8}$ = 4.3 x $10^{-6}$, $p_{URA7}$ = 5.8 x $10^{-6}$) with post hoc Tukey HSD test. (H) Cells of the indicated genotypes expressing endogenous HA-tagged Ura8 were treated with 300 µM DON and fixed in TCA at the indicated time points. Proteins were analyzed by Western blot. The adjacent scheme summarizes the nucleotide alterations induced by *URA7* deletion. (I) Exponentially growing cells of the indicated genotypes in the W303 background were treated with 300 µM DON for 2h as indicated. Metabolites were extracted and NTP levels were quantified by nano-LC-MS/MS. Values are ion intensities (peak areas) normalized by the mean of untreated wt samples. Significance was calculated by Student's t-test (two-sided). ND = not detectable (Wilcoxon signed rank exact test for lower level than wt+DON, $p$ = 0.031). (J) Cells of the indicated genotypes in the W303 background were transfected with plasmids encoding GST-tagged URA7, URA8 or GST tag alone, under control of the galactose-inducible GAL1/10 promoter. Cells were cultured for 18h in YP medium with 2% galactose and fixed in TCA. Proteins were analyzed by Western blot. (K) Cells from (J) of the indicated genotypes were adjusted to 2% galactose in YP medium for 24 hours. $10^7$ cells / mL were serially diluted (1:6), spotted on YP + 2% galactose plates with the indicated concentrations of DON and grown for 3d. (L) Model of the contribution of Ura7 and Ura8 protein levels to CTP synthase activity before and during DON exposure. Ura7 is the major expressed CTP synthase in untreated cells. DON inhibits CTP synthases irreversibly and induces the production of Ura8, which then becomes the major expressed CTP synthase. Ura7 and Ura8 are hence constitutive and inducible CTP synthases, respectively. In *ura7Δ* mutants, Ura8 is already induced in the absence of DON. In *ura8Δ* mutants, Ura7 is inactivated by DON and the absence of an inducible CTP synthase causes a critically low CTP synthase activity. The model is simplified and posttranslational and allosteric regulations further contribute to CTP synthase regulation. Bars plots with error bars represent mean values and standard deviation. Representative Western blot and spot assay images are shown. Pgk1 was used as loading control in Western blots. wt = wildtype, EV = empty vector, Untr = Untreated, ND = not detectable.

*ura8Δ*), *rtg1Δ* and *hpt1Δ* mutants were clearly sensitive to HU, whereas *ura8Δ* mutants were exclusively sensitive to DON but not HU (Fig 1B). This may reflect their different contribution to purine metabolism (and hence dATP production), but could also imply a specific function of Ura8 in the response to DON. We investigated the genetic interactions of *URA8* with *HPT1* and *RTG1* by analyzing the sensitivity of double mutants to DON. The *ura8Δrtg1Δ* mutant displayed synergistically enhanced DON sensitivity (Fig 1C), establishing Rtg1 and Ura8 as genetically separate glutamine analog resistance factors involved in glutamine metabolism and CTP synthesis, respectively (Fig 1D). In contrast, the *ura8Δhpt1Δ* mutant was only as DON-sensitive as the *hpt1Δ* mutant alone (Fig 1C), implying that their role in mediating DON resistance is based on their common activity in nucleotide metabolism (Fig 1D). Hence, disrupting the supply of a single or several nucleotides has a similar impact on glutamine analog sensitization.

Intriguingly, our screen identified only the minor CTP synthase (Ura8) as DON resistance factor, but not the major CTP synthase (Ura7), which is expressed two-fold higher than Ura8 [31] (S1 Table). We confirmed the selective DON sensitivity of *ura8Δ* vs. *ura7Δ* mutants in the W303 genetic background [32] (Fig 1E). We hypothesized that a specific regulation of Ura8 in response to glutamine analogs may account for the selective sensitivity of its mutant to DON. To test this hypothesis, we engineered the endogenous *URA7* and *URA8* loci by C-terminal haemagglutinin (HA) tag fusion, and then monitored the levels of Ura7-HA and Ura8-HA after G1 arrest by α-factor and release into S phase in the presence or absence of 300 µM DON by Western blotting. DON-treated cells accumulated Ura8-HA, but not Ura7-HA, within two hours after G1 release (Fig 1F). In contrast, Ura8-HA levels remained constant in untreated cells, showing that its expression was not regulated throughout a normal cell cycle. We asked if the *URA8* gene was regulated transcriptionally and measured its mRNA levels by reverse transcription-quantitative PCR (RT-qPCR) after DON exposure. Indeed, *URA8* but not *URA7* mRNA abundance strongly increased within two hours of DON treatment (Fig 1G). Since DON inhibits multiple enzymes besides CTP synthase, we asked if CTP synthase inhibition

was sufficient to induce Ura8. Indeed, genetic ablation of *URA7*, which mimics a specific reduction of CTP synthase activity and reduces the intracellular concentration of CTP but not the other NTPs [33], strongly raised Ura8 expression and prevented its further induction by DON (Fig 1H). To corroborate the role of Ura8 in CTP pool maintenance during DON exposure, we measured NTP levels in wt and *ura8Δ* cells with and without DON treatment for 2h. DON caused a more than two-fold increase of intracellular glutamine (S1C Fig), consistent with a broad inhibition of glutamine-dependent reactions and the consequent accumulation of the reaction substrate, glutamine. DON also depleted CTP more than 10-fold in wt cells, whereas UTP and ATP were not affected and there was a trend for reduced GTP levels (Fig 1I). In *ura8Δ* cells, CTP was undetectable after DON treatment in all replicates and hence lower than in the DON-treated wt (Fig 1I), supporting the idea that Ura8 ameliorated CTP depletion by DON. In summary, Ura8 is an inducible CTP synthase which counteracts the inhibition of CTP synthesis, whereas Ura7 is a constitutive CTP synthase that is not involved in this adaptive response.

An intrinsic resistance of the Ura8 enzyme towards DON inhibition could contribute to its specific requirement for survival during DON exposure; a similar mechanism has been previously described for the IMP dehydrogenase Imd2 which confers resistance to the IMP dehydrogenase inhibitor mycophenolic acid, through a combination of its gene inducibility and lower enzyme inhibition in comparison to other IMP dehydrogenase isoforms [34]. We therefore tested if Ura7 and Ura8 had different capacities to restore DON resistance when overexpressed at comparable levels in *ura8Δ* cells. We achieved similar expression levels of Ura7 and Ura8 (Fig 1J), and we found that both CTP synthases similarly restored DON resistance to wt levels (Fig 1K). This observation suggests that likely Ura8 inducibility rather than an intrinsic property of Ura8 accounts for its role in DON resistance, although a characterization of Ura8 and Ura7 enzymatic activities during DON treatment would be required to fully assess inherent inhibition resistance. Since DON is an irreversible CTP synthase inhibitor, our data suggest that newly produced Ura8 constitutes the majority of active CTP synthase under DON exposure and thereby confers resistance to the glutamine analog (Fig 1L). We also found that Ura7 underwent an electrophoretic mobility shift in response to DON (Fig 1F), which may reflect previously described regulatory phosphorylation events [35,36]; however, although the activities of both CTP synthases are known to be controlled at posttranslational and allosteric levels, the exclusive expression control of Ura8 provides a simple explanation for its specific requirement as glutamine analog resistance factor (Fig 1L).

## Hpt1 maintains ATP and GTP balance during DON treatment

We hypothesized that *hpt1Δ* mutants should show disturbed purine metabolism, and measured NTP levels in wt and *hpt1Δ* cells before and after DON treatment for 2h. Indeed, *HPT1* deletion strongly reduced ATP and GTP levels during DON treatment (S1D Fig). This was accompanied by an induction of the GTP-repressible *IMD2* gene (S1E Fig). Together, this supports the idea that Hpt1 is required during DON exposure to maintain purine NTP levels.

## Induction of Ura8 mediates resistance to inhibition of glutamine synthesis

Similar to general inhibition of glutamine metabolism, low glutamine availability could also reduce intracellular CTP levels. We therefore investigated if Ura8 was also involved in the response to glutamine depletion. We cultured cells in a glutamine-free minimal medium and treated them with the glutamine synthetase inhibitor Methionine sulfoximine (MSX), which suppresses the conversion of glutamate to glutamine (Fig 2A). Similar to DON, MSX rapidly induced Ura8 expression (Fig 2A), suggesting that cells respond to glutamine limitation by

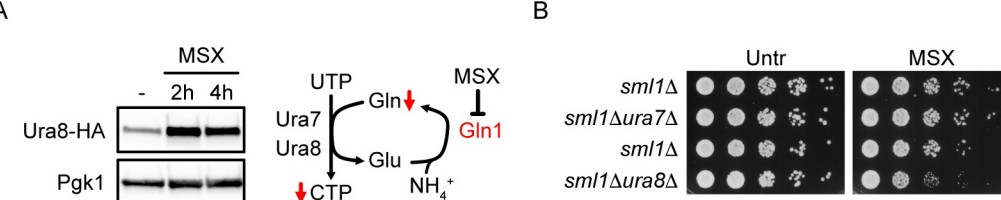

**Fig 2. Glutamine synthetase inhibition induces Ura8.** (A) Cells expressing endogenous HA-tagged Ura8 were cultured in synthetic minimal medium, treated with 15 μM MSX and fixed in TCA at the indicated time points. Proteins were analyzed by Western blot. The adjacent scheme summarizes the expected metabolite alterations induced by MSX. (B) $10^7$ cells / mL of the indicated genotype in the W303 background were serially diluted (1:6), spotted on synthetic minimal media plates with 2% glucose, with or without 15 μM MSX and grown for 2d. Representative Western blot and spot assay images are shown. Gln = glutamine, Glu = glutamate, wt = wildtype, Untr = Untreated.

increasing CTP synthase activity through Ura8 expression. In contrast to DON, MSX does not directly or irreversibly inactivate CTP synthase, raising the question if CTP synthase activity was limiting for cell survival during MSX treatment. We measured the sensitivity of constitutive (*ura7Δ*) and inducible (*ura8Δ*) CTP synthase mutants to MSX and found that specifically *ura8Δ* mutants were sensitive to MSX (Fig 2B). Hence, Ura8 acts as crucial inducible CTP synthase during both glutamine mimetic and glutamine depletion treatments.

## The Mec1-Rad53 DNA damage response pathway mediates DON resistance but not Ura8 regulation

The Mec1[ATR]-Rad53[CHK1/CHK2] pathway senses dNTPs at progressing replication forks and coordinates various cellular processes relevant for DNA replication in response to dNTP shortage. The reduced CTP synthase activity in the *ura7Δ* mutant has previously been shown to activate the Mec1-Rad53 pathway by depletion of CTP and hence dCTP levels [33]. We therefore hypothesized that Mec1 could be activated by DON through inhibition of NTP and dNTP metabolism and orchestrate the glutamine analog response to regulate *URA8* expression. To test this hypothesis, we first monitored phosphorylation of the DNA damage response (DDR) kinase Rad53, the direct target of the apical kinases Mec1 and Tel1[ATM], along with Ura8 levels and cell cycle progression in the presence and absence of DON after release from alpha-factor G1 arrest. Untreated cells progressed through S phase within 60–70 minutes after alpha factor release (Fig 3A). DON treatment did not affect bulk S phase progression, but caused S phase retention of a fraction of cells (Fig 3A). While Rad53 remained unphosphorylated in untreated cells, DON progressively induced Rad53 phosphorylation from 60–70 minutes onwards after alpha factor release, coinciding with S phase completion and the induction of Ura8-HA (Fig 3B). We measured the transcriptional DDR target Rrn3 as readout of Rad53 activity and found that its expression was induced by DON (Fig 3C). To distinguish between Mec1 and Tel1 activation, we then monitored the DON response in *sml1Δmec1Δ* cells, which lack the essential apical replication stress response kinase Mec1 but retain viability by disruption of the physiological Mec1 target and ribonucleotide reductase (RNR) inhibitor Sml1 [37]. Rad53 phosphorylation and Rrn3 induction were nearly absent in DON-treated *sml1Δmec1Δ* cells in comparison with the *sml1Δ* control (Fig 3D), placing Mec1 upstream of the glutamine analog-induced DDR. Due to the lethality of *mec1Δ* and *rad53Δ* mutants in an *SML1* proficient background, these key DDR genes were not covered by our initial DON sensitivity screen. We therefore analyzed the DON sensitivity of *sml1Δ*, *sml1Δmec1Δ*, *sml1Δrad53Δ* and *sml1Δrad53Δhht2Δ* cells by spot assay. Deletion of *HHT2*, encoding a copy of histone H3, restores normal growth of *sml1Δrad53Δ* mutants by compensating for a histone turnover

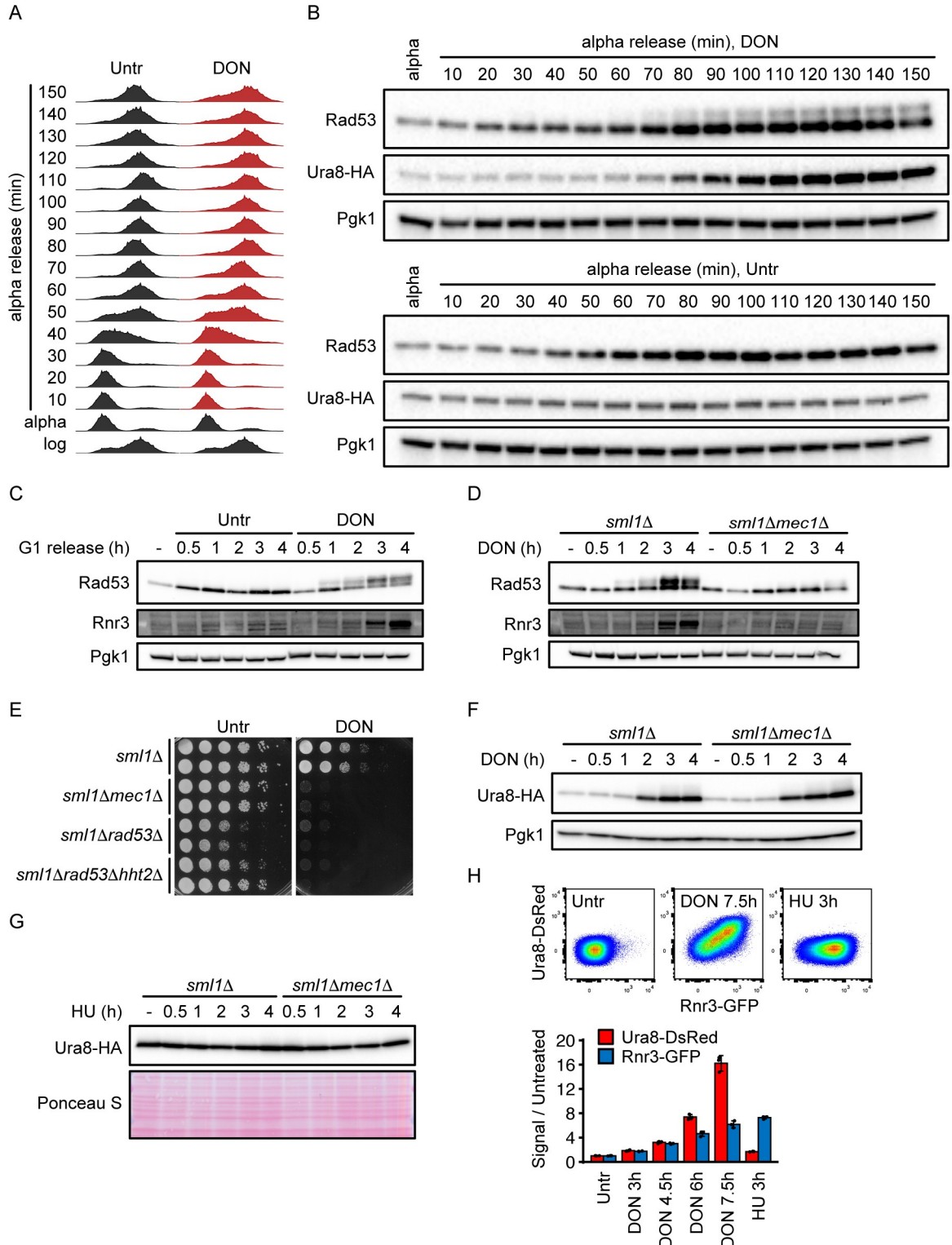

**Fig 3. DDR and CTP synthase mediate independent layers of the glutamine analog response.** (A) Cells of the indicated genotypes were arrested in G1 phase with α-factor, released into YPD with or without 300 μM DON and fixed in 70% ethanol at the indicated time points. DNA content was analyzed by flow cytometry. (B-C) Cells expressing endogenous HA-tagged Ura8 (B) or wildtype cells (C) were arrested in G1 phase with α-factor and released into YPD with or without 300 μM DON. Cells were fixed in TCA at the indicated time points and proteins were analyzed by Western blot. (D) Cells of the indicated genotypes were arrested in G1 phase with α-factor, released into YPD with 300 μM DON and fixed in TCA at the indicated time points. Proteins were analyzed by Western blot. (E) $10^7$

cells / mL of the indicated genotype in the W303 background were serially diluted (1:6), spotted on YPD plates with or without 100 μM DON and grown for 2d. (F, G) Cells of the indicated genotypes expressing endogenous HA-tagged Ura8 were arrested in G1 phase with α-factor, released into YPD with 300 μM DON or 200 mM HU and fixed in TCA at the indicated time points. Proteins were analyzed by Western blot. (H) Cells expressing endogenous DsRed-tagged Ura8 and GFP-tagged Rnr3 were exposed to 300 μM DON or 50 mM HU and fixed with formaldehyde at the indicated time points. Expression of the tagged proteins was analyzed by flow cytometry. The bar plot shows the sample means across 3 independent replicate cultures. Bars plots with error bars represent mean values and standard deviation. Representative Western blot and spot assay images are shown. Pgk1 and Ponceau S were used as loading controls in Western blots as indicated. Untr = Untreated.

defect [38,39]. Indeed, *sml1Δmec1Δ*, *sml1Δrad53Δ* and *sml1Δrad53Δhht2Δ* mutants were hypersensitive to DON (150 μM) (Fig 3E), supporting a role of the Mec1-Rad53 pathway in the glutamine analog response.

To investigate a possible regulation of *URA8* by Mec1, we constructed *sml1Δmec1Δ* and *sml1Δ* cells with HA-tagged endogenous Ura8 and measured its induction after DON treatment. Ura8-HA was induced with equal efficiency in *sml1Δmec1Δ* and *sml1Δ* cells (Fig 3F), suggesting that Mec1 was not an upstream regulator of Ura8. Consistently, strong stimulation of Mec1 activity by HU did not increase Ura8-HA levels (Fig 3G). To corroborate this finding, we constructed a reporter strain in which the endogenous *URA8* gene is tagged with two copies of the fluorophore DsRed, and the endogenous *RNR3* gene with green fluorescent protein (GFP). Flow cytometry analysis confirmed that DON efficiently induced both Ura8-DsRed and Rnr3-GFP in the reporter strain, whereas HU mainly induced Rnr3-GFP (Fig 3H). Hence, the Mec1-Rad53 pathway mediates an essential DDR to the glutamine analog DON, which does not control Ura8 induction.

## CTP synthase Ura8 limits DDR activation in response to DON

Our data support a model in which the inducible CTP synthase Ura8 acts upstream of the DDR in response to glutamine analogs to prevent replication stress. This model predicts that disrupting nucleotide homeostasis should increase the requirement for a functional DDR during glutamine analog exposure. We therefore analyzed the genetic interaction of *MEC1* with the constitutive (*URA7*) and inducible (*URA8*) CTP synthase genes by tetrad dissection. We found that *sml1Δmec1Δura7Δ* but not *sml1Δmec1Δura8Δ* mutants were inviable in the absence of DON (Fig 4A). While the untreated *sml1Δmec1Δura8Δ* mutant had no obvious growth defect, it was extremely sensitive to low doses of DON (50 μM) or MSX (15 μM) (Fig 4B). Hence, disruption of the constitutive CTP synthase (*ura7Δ*) renders cells dependent on Mec1 for survival, whereas disruption of the inducible CTP synthase (*ura8Δ*) increases Mec1 requirement only in combination with glutamine analog treatment or glutamine synthetase inhibition. To further corroborate that CTP synthase becomes limiting for nucleotide homeostasis in DDR mutants under DON exposure, we ectopically expressed *URA7* and *URA8* from plasmids with a galactose-inducible promoter in *sml1Δ* and *sml1Δmec1Δ* cells. Indeed, overexpression of either *URA7* or *URA8* partially suppressed the DON hypersensitivity of *sml1Δmec1Δ* cells (Fig 4C). According to our model, Ura8 should diminish the activity of the Mec1-Rad53 pathway in response to continuous DON exposure. To test this, we monitored Rad53 phosphorylation in DON-treated wt and *ura8Δ* cells. As predicted, a higher fraction of Rad53 was phosphorylated in the *ura8Δ* mutant (Fig 4D). Notably, while wt cells recovered unphosphorylated Rad53 between 4 hours and six hours of treatment, nearly the entire Rad53 pool remained phosphorylated in the *ura8Δ* mutant (Fig 4D). Together, these data support that CTP synthase Ura8 prevents DNA damage and reduces the requirement for a DDR during inhibition of glutamine metabolism.

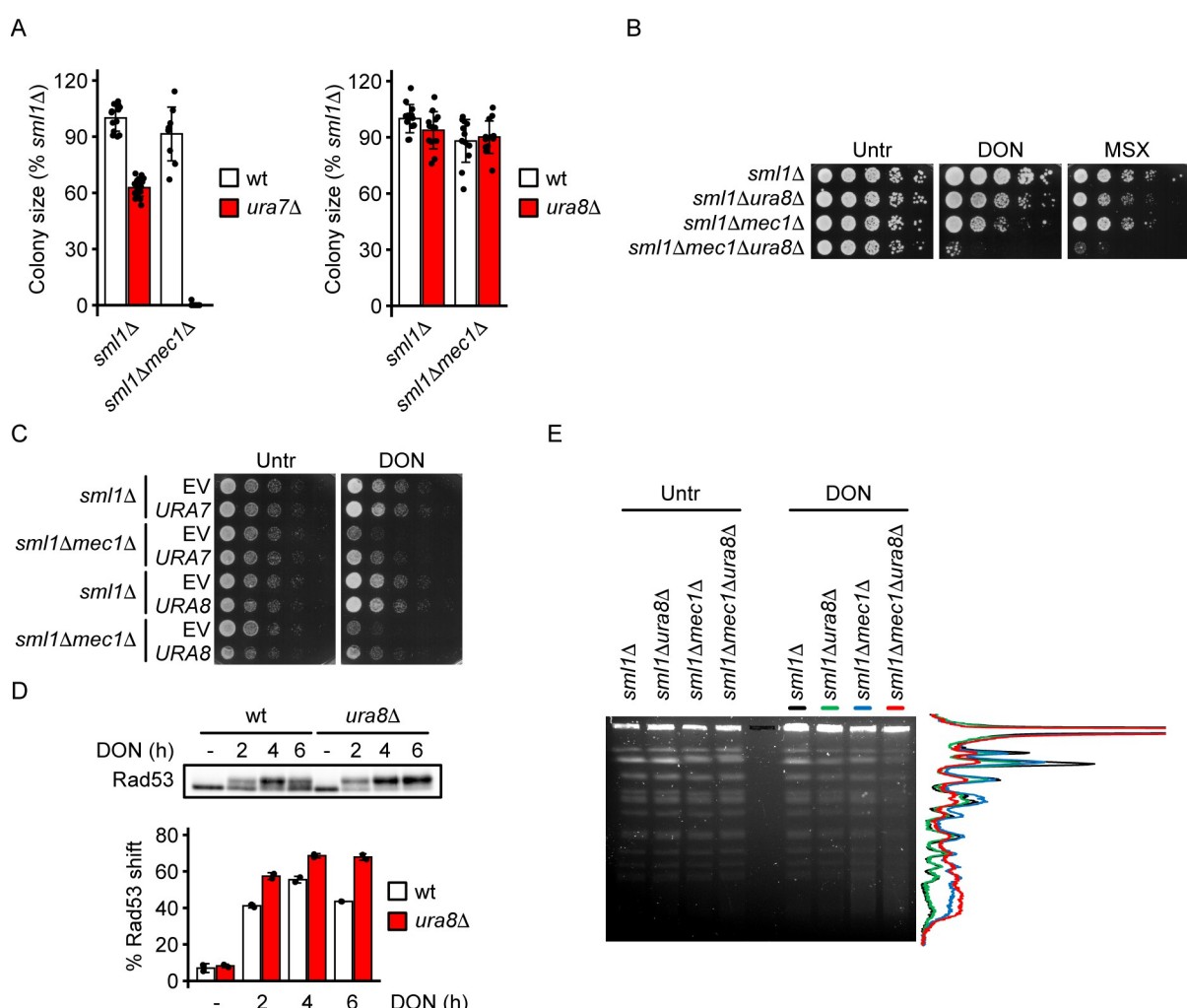

**Fig 4. CTP synthase limits DDR activation by glutamine analogs.** (A) Analysis of the genetic interaction of *MEC1* with *URA7* (left panel) and *URA8* (right panel) by tetrad dissection. Tetrads were dissected into single spores, colony sizes were measured after 3 days at 28˚C and genotypes were determined by replica plating. (B) $10^7$ cells / mL of the indicated genotype in the W303 background were serially diluted (1:6), spotted on YPD plates with 50 µM DON or 15 µM MSX as indicated and grown for 2d. (C) $10^7$ cells / mL of the indicated genotype in the W303 background expressing *URA7* or *URA8* from a plasmid with a galactose-inducible promoter were serially diluted (1:6), spotted on YP + 2% galactose plates with or without 150 µM DON and grown for 3d. (D) Cells of the indicated genotypes were arrested in G1 phase with α-factor, released into YPD with 300 µM DON and fixed in TCA at the indicated time points. The electrophoretic mobility shift of the Rad53 protein reflecting its phosphorylation was analyzed by Western blot. The percentage of phosphorylated vs. total Rad53 was quantified using ImageJ software and represented in the bar plot (n = 2 independent replicate cultures). (E) Cells of the indicated genotypes were arrested in G1 phase with α-factor, released into YPD with or without 50 µM DON for 6h and fixed with sodium azide. Chromosome integrity was analyzed by pulsed-field gel electrophoresis analysis and staining of DNA with ethidium bromide. The right panel shows the quantification of DNA signal along the gel lanes containing DON-treated samples. Bars plots with error bars represent mean values and standard deviation. Representative Western blot, DNA gel and spot assay images are shown. wt = wildtype, EV = empty vector, Untr = Untreated.

The Mec1-Rad53 pathway suppresses chromosome breakage during dNTP shortage [22]. We asked if Mec1 also prevented chromosome breakage in response to DON. We cultured the *sml1Δ*, *sml1Δmec1Δ*, *sml1Δura8Δ* and *sml1Δmec1Δura8Δ* cells in a dose of DON which causes synergistic lethality of *sml1Δmec1Δura8Δ* mutants (see Fig 4B), and analyzed chromosome fragmentation by pulsed-field gel electrophoresis (PFGE) after 6 hours. While chromosome integrity was not visibly affected in untreated cells in any of the tested genotypes, DON specifically induced chromosome breaks in *sml1Δmec1Δ* and *sml1Δmec1Δura8Δ* but not *sml1Δ* or

*sml1Δura8Δ* cells, as shown by a diffuse DNA signal between and below the lower molecular weight chromosome bands in the PFGE (Fig 4E). We did not observe a synergistic increase of chromosome breakage in *sml1Δmec1Δura8Δ* vs. *sml1Δmec1Δ* cells, implying that a potential impact of Ura8 on chromosome stability through CTP synthesis could become relevant during long-term DON exposure. In summary, Mec1 suppresses chromosome breakage in response to DON.

## Ura8 induction is specific to pyrimidine limitation

Our data show that Ura8 induction is not regulated by the DDR. We therefore asked how the inhibition of glutamine metabolism was coupled to Ura8 induction mechanistically. Since DON and *URA7* deletion specifically reduce CTP and induce Ura8, we hypothesized that CTP (or CTP-derived metabolites) may act as signal for Ura8 induction. This hypothesis predicts that limitation of pyrimidine (UTP, CTP) but not purine (ATP, GTP) nucleotides should induce Ura8. Pyrimidine nucleotides are synthesized from exogenous Uracil through the salvage pathway in the yeast W303 (and S288C) background. We therefore applied a Uracil limitation range which impairs the production of both CTP and UTP (Fig 5A). To control for the capacity of Uracil-restricted cells to perform inducible transcription of the *URA8* gene, we included an MSX treatment for each step of Uracil limitation. Indeed, cells at the highest Uracil concentration induced Ura8 efficiently after MSX, whereas a severe Uracil limitation (256-fold) prevented its induction (Fig 5A). A less stringent but transcription-permissive Uracil limitation (64-fold) caused a strong induction of Ura8 in the absence of MSX (Fig 5A). In contrast, Ura8 was not responsive to Adenine starvation in the auxotrophic W303 strain, which limits ATP and GTP production (Fig 5B). We conclude that the induction of Ura8 can be achieved by CTP synthase inhibition, glutamine synthetase inhibition or pyrimidine base starvation but not by depletion of purine nucleotides. These results are consistent with CTP (or CTP-derived metabolites) serving as signal for Ura8 regulation.

## Ura8 regulation involves the Nrd1/Nab3 complex and a cryptic unstable transcript

Enzymes catalyzing rate-limiting steps in nucleotide synthesis are regulated by nucleotide levels. Such regulations are mediated by the activity of trans-acting transcription factors [40], and by cis-acting small upstream DNA regions encoding cryptic unstable transcripts (CUTs). These CUTs are produced when NTPs are abundant, and their production involves transcription termination through the Nrd1/Nab3 complex which prevents progression of RNA polymerase II into the main ORF [41–45].

We analyzed by RT-qPCR how DON affected the expression of the main NTP-responsive genes. The trans-regulated UTP-responsive gene *URA1* was not induced by DON, supporting a specific CTP response and not a general pyrimidine response (Fig 5C). DON de-repressed the CUT-regulated purine metabolism genes *IMD2* and *ADE12* (Fig 5C), albeit to a lesser extent than *URA8*, reflecting a minor CUT-mediated purine starvation response. The CUT-regulated *URA2* gene was only mildly and transiently induced (Fig 5C). Hence, DON triggers CUT-regulated CTP and purine synthesis genes. This is consistent with the direct inhibition of CTP synthase and purine *de novo* synthesis enzymes by DON [4].

The *URA8* gene is preceded by a CUT and is spontaneously induced by RNA polymerase II mutations simulating general lack of NTPs [41,44]. The region between the CUT transcription start site and the main ORF, termed R box, contains five binding sites for the transcription-terminating Nrd1/Nab3 complex (Fig 5D) which mediates CUT transcription termination at the *IMD2* and *URA2* loci in the presence of GTP or UTP, respectively [43,44]. In contrast to other

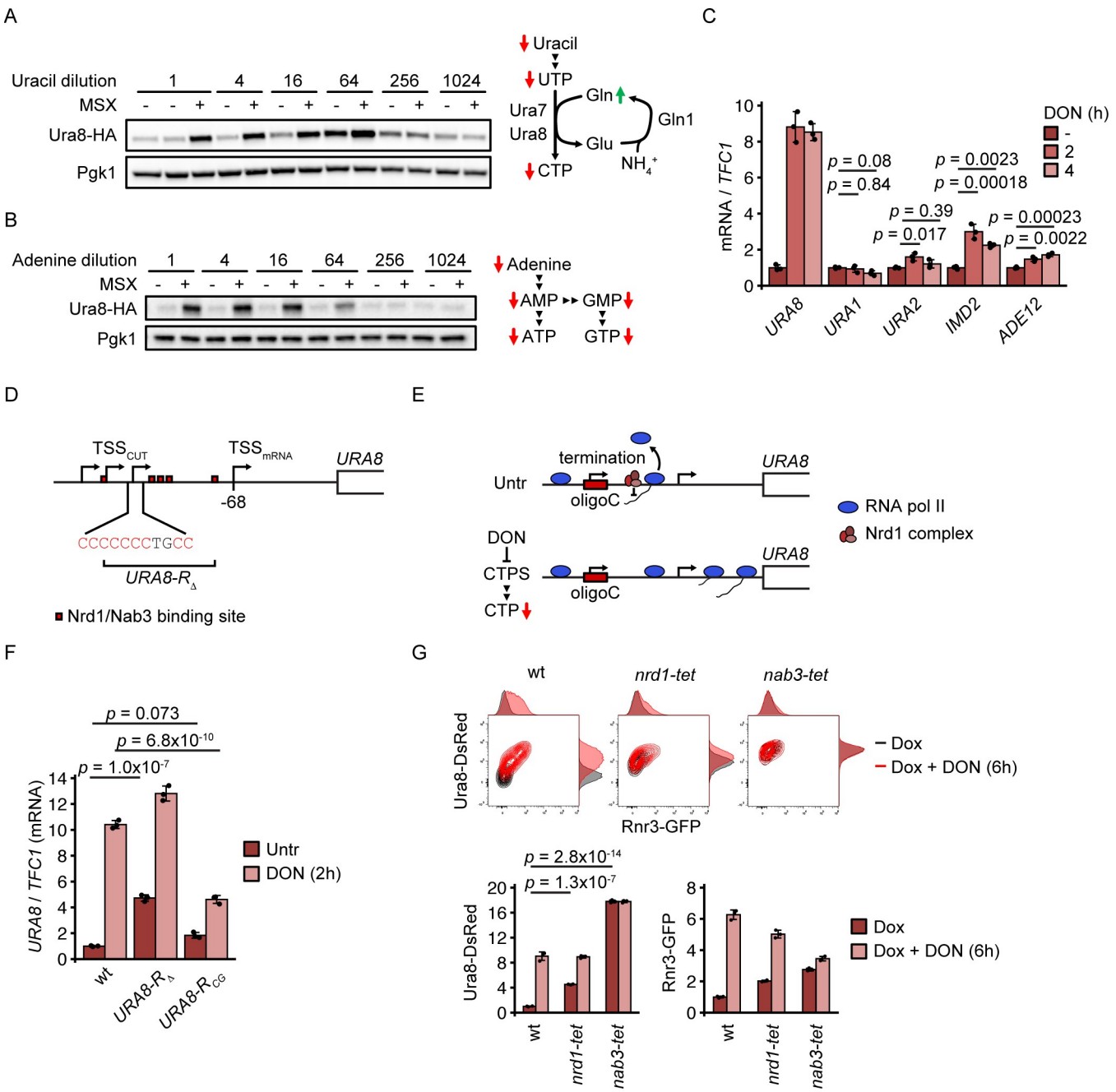

**Fig 5. Regulation of the *URA8* locus in response to glutamine analogs.** (A, B) Cells expressing endogenous HA-tagged Ura8 were cultured in synthetic minimal medium, transferred to medium with the indicated dilutions of Uracil (A) or Adenine (B), with or without 15 μM MSX, and fixed in TCA after 4h. Proteins were analyzed by Western blot. The adjacent schemes summarize the expected metabolite alterations induced by restriction of Uracil (A) or Adenine (B). (C) Wildtype cells grown in YPD medium were treated with 300 μM DON for the indicated duration. Samples were collected to quantify the indicated mRNAs by RT-qPCR using *TFC1* as reference (n = 3 independent replicate cultures). Significances were calculated with 1-way ANOVA ($p_{URA1}$ = 0.0781, $p_{URA3}$ = 0.0119, $p_{URA2}$ = 0.0194, $p_{IMD2}$ = 0.000212, $p_{ADE12}$ = 0.000265) with post hoc Tukey HSD test. The same samples as in Fig 1G were used, and the *URA8* expression data from Fig 1G was included for quantitative comparison. (D) Schematic of the *URA8* ORF upstream DNA region. Information on transcription start sites is included. Oligo-C tract and R box region modified in delitto perfetto mutants are indicated. The coding strand DNA sequence is represented. (E) Working model of *URA8* regulation by transcription start site selection. Top panel, untreated cells: RNA polymerase II initiates transcription at C-containing start sites. The Nrd1 complex terminates transcription leading to CUT production and preventing the progression of RNA polymerase II into the *URA8* ORF. Bottom panel: Low CTP levels suppress transcription initiation at C-containing start sites. RNA polymerase II initiates transcription downstream of Nrd1 complex binding sites and transcribes the *URA8* ORF. (F) Delitto perfetto mutants with *URA8* R box deletion (*URA8-R$_\Delta$*) or C-to-G substitutions in the *URA8* upstream oligo-C tract shown in (D) (*URA8-R$_{CG}$*) were arrested in G1 phase with α-factor and released into YPD with or without 300 μM DON for 2h. Samples were collected to quantify *URA8* mRNA by RT-qPCR using *TFC1* as reference (n = 3 independent delitto perfetto clones). Significances were

calculated with 1-way ANOVA ($1.43 \times 10^{-7}$) with post hoc Tukey HSD test. (G) Strains with doxycycline-repressible alleles of the Nrd1 complex (*nrd1-tet*, *nab3-tet*) from the Tet-Promoters Hughes library were crossed with a query strain expressing endogenous DsRed-tagged Ura8 and GFP-tagged Rnr3. Mutant offspring was cultured overnight in YPD with 50 μM doxycycline, treated with 300 μM DON in YPD with 50 μM doxycycline and fixed with formaldehyde after 6h. Expression of the tagged proteins was analyzed by flow cytometry (n = 3 independent replicate cultures). Significances were calculated with 1-way ANOVA ($p_{Ura8}$ = $3.47 \times 10^{-16}$) with post hoc Tukey HSD test. Bars plots with error bars represent mean values and standard deviation. Representative Western blot and spot assay images are shown. Gln = glutamine, Glu = glutamate, wt = wildtype, Untr = Untreated.

genes regulated by an R box (*IMD2*, *URA2*, *ADE12*), the *URA8* R box contained an oligo-C sequence immediately upstream of a Nrd1/Nab3 binding site cluster. This sequence coincides with a CUT transcription initiation site [44] (Fig 5D). CUT transcription initiation at this sequence would require CTP as initiating nucleotide. Low CTP levels could hence prevent CUT initiation and allow the progression of RNA polymerase II into the *URA8* ORF by avoiding Nrd1/Nab3-mediated transcription termination (Fig 5E). The same mechanism of NTP-dependent start site selection has previously been shown to couple GTP levels to *IMD2* expression [43]. We therefore hypothesized that the Nrd1/Nab3 complex and the CUT regulate *URA8* expression in analogy to the *IMD2* locus, with the difference that the oligo-C sequence facilitates specific CTP sensing (Fig 5E). This hypothesis predicts that disruption of the R box and the Nrd1/Nab3 complex should 1) constitutively de-repress *URA8* and 2) show positive epistatic interactions with DON in *URA8* expression.

We first tested if the R box regulated *URA8* expression and deleted it in a haploid strain with the marker-free delitto perfetto technique [46] (*URA8-R$_\Delta$*, Fig 5D). We measured *URA8* mRNA levels before and after DON treatment by RT-qPCR and found that untreated *URA8-R$_\Delta$* cells spontaneously de-repressed *URA8* (Fig 5F). Importantly, DON induced *URA8* mRNA to similar absolute levels in wt and *URA8-R$_\Delta$* cells (Fig 5F), suggesting that the R box exerts a repressive effect on *URA8* transcription which is alleviated by DON treatment. We next tested the involvement of the Nrd1/Nab3 complex in the regulation of Ura8 induction after DON treatment. Since all Nrd1 complex components (Nrd1, Nab3, Sen1) are essential, we crossed strains from the Tet-Promoters Hughes library (Tet-Off) [47] for doxycycline-controlled repression of the *NRD1* and *NAB3* genes with the Ura8-DsRed Rnr3-GFP reporter strain. We depleted the expression of *NRD1* and *NAB3* with doxycycline and measured Ura8 and Rnr3 expression with and without DON exposure. Depletion of either Nrd1 complex component induced Ura8 in the absence of DON (Fig 5G). DON elevated the expression of Ura8 to the same level in wt and Nrd1-depleted cells but induced the DDR activation marker Rnr3-GFP less in Nrd1-depleted cells than in wt (Fig 5G). Nab3-depleted cells were severely slow-growing and showed an overall stronger Ura8 and weaker Rnr3 induction (Fig 5G). We confirmed that the observed differences in Ura8 levels were not a consequence of altered cell size (S2A Fig). These observations are in agreement with a previous study showing that CUT transcription termination defects at the *URA8* CUT under condition of abundant NTPs are more severe in *nab3* mutants than in *nrd1* mutants [45].

We decided to investigate if the oligo-C sequence was mechanistically involved in *URA8* expression. We performed a C-to-G substitution with the delitto perfetto method (*URA8-R$_{CG}$*) which preserves the GC content but removes the C from one of the CUT initiation sites. The *URA8-R$_{CG}$* mutation caused a mild spontaneous de-repression of the *URA8* locus (1.8-fold) (Fig 5F), consistent with *URA8* repression by transcription initiation at the CUT start site. However, the mutation also strongly reduced the inducibility of *URA8* in response to DON (Fig 5F). Thus, our data are in agreement with start site selection as regulatory mechanism of *URA8* expression, but also suggest a role of the upstream oligo-C tract in promoting the efficient transcription of *URA8* during DON treatment, which we cannot explain entirely by start site selection (see Discussion).

In summary, our data suggest that 1) the Nrd1/Nab3 complex and the upstream CUT repress the constitutive expression of *URA8*, similar to *IMD2* and *URA2* regulation [43,44] and 2) an oligo-C sequence may couple CTP levels to *URA8* transcription. Our data together with previous works reinforce the idea that the intracellular levels of NTPs are sensed at the level of RNA transcription.

## TORC1, glutamine transporters and transcription regulators modulate glutamine analog sensitivity of DDR mutants

The *sml1Δmec1Δ* mutant is hypersensitive to CTP synthase inhibition while executing a normal upstream glutamine analog response. It is therefore a suitable model to screen for mechanisms of glutamine analog resistance. We screened the YKO library for suppressors of the *sml1Δmec1Δ* DON sensitivity and identified 67 suppressors with stringent criteria (FDR = 10%, $\log_2$ relative colony size DON vs. untreated > 0.5) encoding for 61 annotated genes and 6 dubious open reading frames of which 5 overlapped with candidate genes (Fig 6A and S3 Table). We stratified these hits by scoring for HU sensitivity (Figs 6B and S3A and S3 Table) and suppression of the *ura8Δ* DON sensitivity in secondary screens (Figs 6B and S3B and S4 Table). Gene network visualization by STRING [48] revealed that most candidates were integrated into three separate clusters: A transport and metabolism cluster (Fig 6B left side), a cluster related to chromatin and transcription (Fig 6B, right side), and a cluster of Elongator complex components involved in RNA polymerase II progression and tRNA biogenesis (Fig 6B, middle bottom). Nearly all candidates suppressed the sensitivity of both *sml1Δmec1Δ* and *ura8Δ* mutants with few exceptions (*HDA3*, *CSA1*, *TIM18*, *RPL8A*, *RPL34B*, *ELO3*). Most candidates were specific for DON, and we observed HU resistance mainly in the Elongator and few other chromatin-related candidates (SWR1, Prefoldin). The strongest DON suppression was achieved by the deletion of amino acid transporters (*GNP1*, *TAT2)*, positive regulators of TORC1 signaling from the SEACAT (*MTC5*), EGO/GSE (*GTR1*, *SLM4*, *MEH1*) and Lst4-Lst7 (*LST4*) complexes, and genes involved in transcription regulation such as components of PAF1, prefoldin and elongator complexes (Fig 6B).

The genes influencing glutamine analog hypersensitivity could act at two levels: The stabilization of nucleotide pools or the prevention and handling of replication stress. To distinguish between these mechanisms, we characterized activation of the CTP synthase response and DDR in the candidates using the Ura8-DsRed Rnr3-GFP reporter system (see Fig 3H). Impaired DON import or enhanced glutamine availability are expected to diminish Ura8 and Rnr3 induction, whereas reduced replication stress should specifically reduce Rnr3 induction. Several mutants with reduced TORC1 activity or altered amino acid transporter levels, localization or turnover, showed reduced Ura8-DsRed and Rnr3-GFP induction (Fig 6C). In contrast, mutants of the RNA polymerase II phosphatase and the PAF1, SWR1, prefoldin and elongator complexes responded to DON with Ura8-DsRed induction, but expressed less Rnr3-GFP, indicative of reduced DDR activation (Fig 6C). Hence, these mutations are independent from CTP synthase induction and reduce the DDR requirement.

To corroborate the potential roles of amino acid transporters and TORC1 activators in CTP depletion by DON, we decided to analyze the impact of *GNP1* and *GTR1* deletion on CTP levels. We confirmed that deletion of *GNP1* and *GTR1* suppressed the DON but not HU sensitivity of *sml1Δmec1Δ* mutants (Fig 7A). Moreover, deletion of *GNP1* and *GTR1* ameliorated CTP depletion (Fig 7B), further suggesting an involvement of the glutamine transporter Gnp1 and TORC1 activity in targeting of CTP metabolism by DON.

RNA polymerase II phosphatase and the PAF1, elongator, and prefoldin complexes act on the chromatin and may therefore mediate the chromosome instability of *sml1Δmec1Δ* mutants

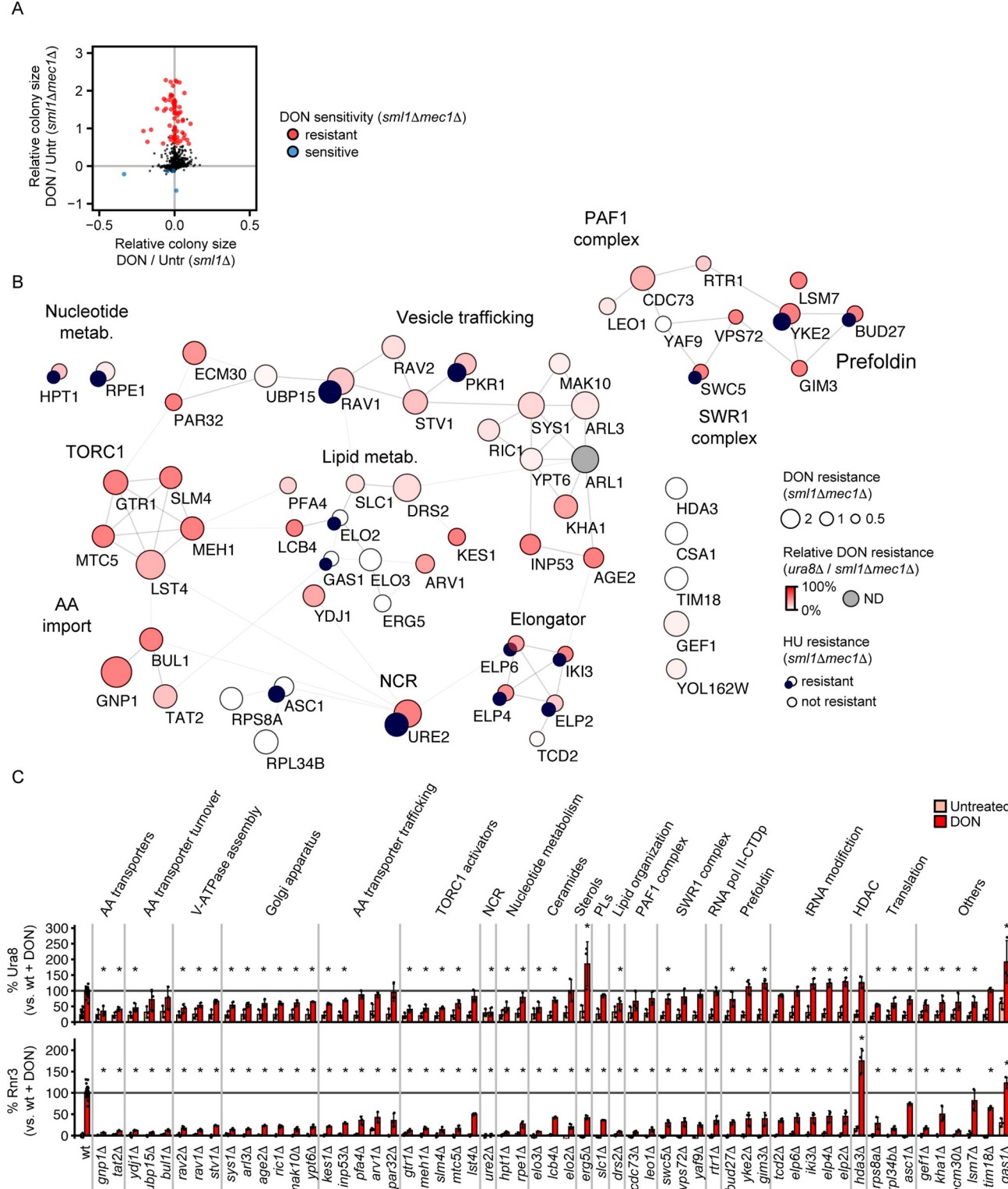

**Fig 6. Genetic suppression of glutamine analog hypersensitivity.** (A) The viable haploid gene deletion yeast library was crossed with *sml1Δ* and *sml1Δmec1Δ* query strains. Mutant offspring was selected, replicated on YPD (Untreated) or YPD DON, and colony sizes were quantified. Each dot represents a library mutant. The x and y axes represent the relative colony sizes in the *sml1Δ* (x) and *sml1Δmec1Δ* (y) backgrounds on YPD + DON vs. YPD after 70-percentile normalization to all other *sml1Δ* and *sml1Δmec1Δ* colonies, respectively. All values are log₂-transformed and adjusted for standard deviation. Significant sensitivity or resistance to DON in the *sml1Δmec1Δ* background (see Materials and Methods) is indicated in blue

(sensitive) and red (resistant). Statistical information is provided in S3 Table. (B) Clustering of DON hypersensitivity suppressors from (A) with STRING and Cytoscape software. The circle diameter represents the suppressor strength in the *sml1Δmec1Δ* background. The color represents the relative suppressor strength in the *ura8Δ* vs. *sml1Δmec1Δ* background. A small dark blue circle indicates that the library mutation significantly suppresses the HU sensitivity of *sml1Δmec1Δ* mutants. The classifications are manually selected. Statistical information is provided in S3 and S4 Tables. (C) Suppressor candidates from (A) were crossed with a query strain expressing endogenous DsRed-tagged Ura8 and GFP-tagged Rnr3. Mutant offspring was exposed to 300 μM DON and fixed with formaldehyde after 6h. Expression of the tagged proteins was analyzed by flow cytometry (n = 3 independent experiments). Asterisks indicate significance differences between Ura8-DsRed (top panel) and Rnr3-GFP (bottom panel) expression after DON treatment vs. the wt control (Kruskal-Wallis rank sum test followed by two-sided Mann-Whitney test and Benjamini-Hochberg correction, $p_{adj} < 0.1$). Bars plots with error bars represent mean values and standard deviation. wt = wildtype, NCR = nitrogen catabolite repression, ND = not determined.

under CTP depletion. We decided to test this hypothesis by assessing a potential reduction of chromosome breakage in *sml1Δmec1Δ* mutants by deletion of in representative candidates. Deletion of *CDC73* (PAF1 complex), *YKE2* (prefoldin) and *ELP6* (elongator) completely suppressed DON sensitivity of *sml1Δmec1Δ* mutants, whereas deletion of *RTR1* (RNA polymerase II phosphatase) partially suppressed their DON sensitivity (Fig 7C). We confirmed weak

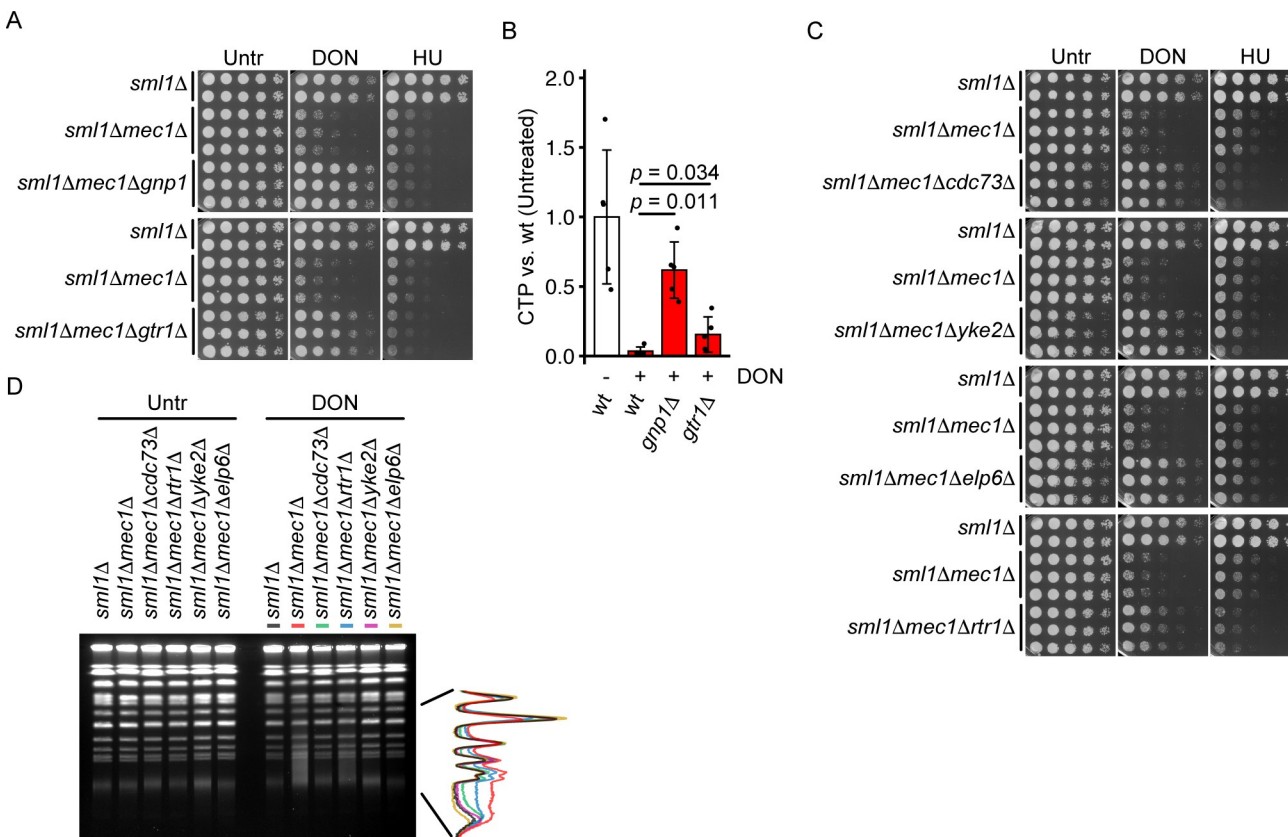

**Fig 7. Validation of glutamine analog hypersensitivity suppressors.** (A) $10^7$ cells / mL of the indicated genotype in the S288C background were serially diluted (1:6), spotted on YPD plates with or without 50 μM DON or 1.5 mM HU and grown for 2d (Untreated, HU) or 3d (DON). (B) Exponentially growing cells of the indicated genotypes in the S288c background were treated with 100 μM DON for 2h as indicated. Metabolites were extracted and CTP levels were quantified by nano-LC-MS/MS. Values are ion intensities (peak areas) normalized by the mean of untreated wt samples. Significance was calculated by Kruskal-Wallis rank sum test ($p = 0.0012$), followed by Wilcoxon rank sum test. (C) $10^7$ cells / mL of the indicated genotype in the S288C background were serially diluted (1:6), spotted on YPD plates with or without 50 μM DON or 1.5 mM HU and grown for 2d (Untreated, HU) or 3d (DON). (D) Cells of the indicated genotypes were arrested in G1 phase with α-factor, released into YPD with or without 50 μM DON for 6h and fixed with sodium azide. Chromosome integrity was analyzed by pulsed-field gel electrophoresis analysis and staining of DNA with ethidium bromide. The left panel shows a representative PFGE result from 2 independent gels. The right panel shows the quantification of DNA signal along the gel lanes containing DON-treated samples. Bars plots with error bars represent mean values and standard deviation. Representative spot assay images are shown. wt = wildtype, NCR = nitrogen catabolite repression, Untr = Untreated.

positive genetic interactions of *MEC1* with *ELP6* (3 clones) and *YKE2* (2 clones) (see Fig 6B); however, deletion of *ELP6* or *YKE2* did not confer a net increase of growth on HU. Deletion of all four candidate genes further suppressed DON-induced chromosome breakage in *sml1Δmec1Δ* mutants as measured by PFGE, with *rtr1Δ* conferring a partial suppression, consistent with its partial effect on DON sensitivity (Fig 7D).

In summary, our data suggest that DON hypersensitivity suppressors can act through at least two mechanisms: shifting the intracellular DON-to-glutamine ratio and reducing DNA break formation. However, we note that the suppression mechanisms are more complex than depicted here. First, not all amino acid transport regulators clearly suppress Ura8 induction (Fig 6C), which could imply additional functions of the candidates independent from CTP regulation. Second, we also identified several candidates belonging to diverse biological processes such as translation, histone deacetylation and lipid metabolism. The role of these candidates in the DON response requires further investigation to better understand the full spectrum of suppression mechanisms.

## Discussion

In this study we applied chemogenomic screens to identify the processes mediating resistance to the glutamine analog DON, and identified CTP synthesis, purine salvage, transcriptional regulation of glutamine metabolism and the replication stress response are the most critical resistance mechanisms (Fig 8). We classify Ura8 as inducible CTP synthase which responds to inhibition of glutamine metabolism and CTP synthesis, and propose a model for the coupling of CTP sensing at the *URA8* locus to rapidly boost CTP synthase expression. Failure to restore CTP synthase activity triggers replication stress and activation of the protective Mec1$^{ATR}$ pathway to avert chromosome breakage. Inactivation of the Mec1 pathway during DON exposure results in chromosome breakage and synergistically enhances the sensitivity of CTP synthase mutants to DON. The impact of DON depends on at least three processes that act upstream of its targets in nucleotide synthesis: Glutamine transporters and TORC1 promote DON activity, whereas the glutamine synthesis-stimulating transcription factor Rtg1 counteracts its toxicity.

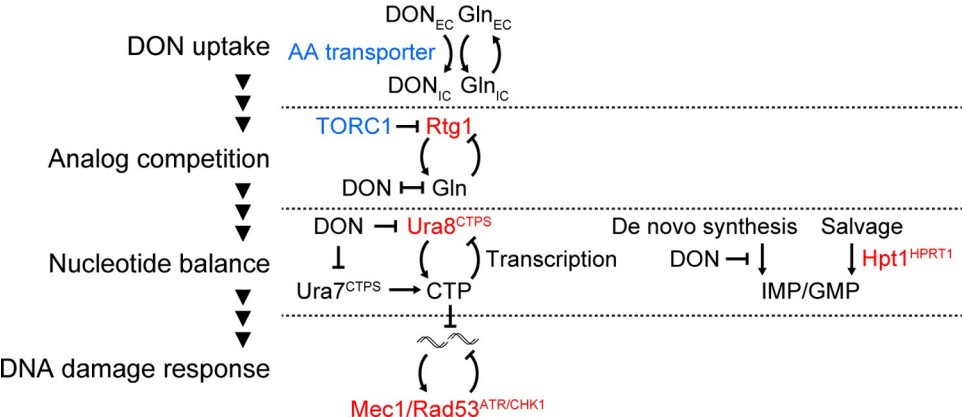

**Fig 8. Hierarchy of pathways determining glutamine analog sensitivity.** Sensitivity towards the glutamine analog DON is modulated by separable pathways at distinct levels: DON toxicity requires amino acid transporters which likely promote its uptake. Intracellularly, TORC1-regulated glutamine synthesis counteracts DON by producing its structural competitor. At the level of nucleotide synthesis, the direct DON target CTP synthase is actively restored through inducible expression of the *URA8* gene to enhance CTP synthesis, whereas purine salvage represents the strategy to counteract purine de novo synthesis blockage. The DNA damage response acts downstream of these mechanisms and prevents DON-induced chromosome breakage. DON resistance and sensitizing genes are shown in red and blue, respectively.

## Ura8 –an inducible CTP synthase responding to inhibition of glutamine metabolism

We show that the glutamine analog DON, which inhibits multiple points of nucleotide metabolism, strongly depletes CTP and thereby increases the requirement for efficient CTP synthesis, which is consistent with the direct and irreversible inhibition of CTP synthase by DON [4]. Although CTP synthesis is not the sole target of this glutamine analog, the intracellular concentration of CTP is the lowest among all nucleotide triphosphates [49], and for the same reason Mec1$^{ATR}$ may be particularly sensitive to CTP synthase inhibition. Consistently, deletion of *URA7*, the gene encoding the constitutive CTP synthase, has been shown to reduce CTP and dCTP levels, increase mutation rate and activate the DDR [33]. CTP synthase is known to be regulated allosterically by ATP, UTP and CTP, through the protein kinases A [36] and C [50], and by the evolutionarily conserved compartmentalization into a filament structure termed cytoophidia [51–53]. Our study adds the transcriptional induction of the minor CTP synthase Ura8 as regulatory mechanism which becomes critical during inhibition of glutamine metabolism and the consequent impairment of CTP synthesis. Since Ura8, but not Ura7, is critical for cell survival to DON, these data imply that Ura8 takes over the role as major CTP synthase during inhibition of glutamine metabolism (Fig 1L). This is consistent with the estimated two-fold higher level of Ura7 in comparison with Ura8 protein [31], contrasted by an approximate 8-fold increase of Ura8 levels shortly after DON treatment.

## Mechanistic coupling of CTP levels to *URA8* expression

Our data show that the *URA8* gene is induced by various conditions that reduce CTP levels specifically (*ura7Δ* mutation) or in combination with other metabolite alterations (DON, MSX, and Uracil starvation), implying a homeostatic feedback loop that enhances CTP synthase activity when CTP levels are low (Fig 8). Differences in enzyme inactivation by DON or different glutamine Km values could further contribute to the different requirement of Ura8 and Ura7 for resistance towards inhibition of glutamine metabolism, and additional biochemical experiment would be required to further investigate these possibilities. The transcription of several key genes encoding NTP synthesis enzymes is directly regulated by NTP levels [41,43,44]. When sufficient GTP is available, transcription of the *IMD2* locus starts at TATA box-proximal "G" sites, producing small upstream transcripts and preventing transcription of the *IMD2* coding sequence. During GTP limitation the selection of downstream non-"G" start sites produces a transcript that contains the *IMD2* ORF, coupling low GTP levels to increased *IMD2* expression [43]. Similarly, the expression of *ADE12*, *URA2* and *URA8* genes, which encode enzymes catalyzing ATP, UTP and CTP synthesis is regulated through the NTP-sensitive production of upstream cryptic unstable transcripts (CUTs) [41,44]. Our study suggests that CTP-sensitive regulation of *URA8* occurs during inhibition of glutamine metabolism by DON and MSX and contributes to the resistance towards these drugs.

## The specificity of CTP sensing at the *URA8* locus

The GTP sensing mechanism at the *IMD2* locus suggests that NTP shortage responses are highly NTP-specific and are not part of a general NTP response: GTP shortage specifically induces *IMD2* expression and therefore enhances GTP synthesis [43]. We show that the *ura7Δ* mutation, which reduces CTP but not UTP, ATP, or GTP levels [33], potently increases Ura8 expression, supporting the idea that reduced CTP levels are sufficient to induce Ura8. By applying a glutamine analog in the presence of uridine for pyrimidine salvage, we achieve a major induction of the *URA8* mRNA (8-fold) but only a comparably minor (1.6-fold) and

transient induction of the *URA2* mRNA, implying separate regulation mechanisms. How could *URA8* expression be specifically coupled to CTP levels? Similarly to *IMD2* regulation, start site selection at "C" sites could account for CTP specificity. In agreement with this mechanism, C-to-G mutation partially de-represses *URA8*, and it is possible that a similar mutation of all "Cs" at the known CUT transcription start sites [44] would result in an even more potent spontaneous *URA8* de-repression. However, we also observe that the C-to-G mutation reduces the DON inducibility of *URA8*, pointing out an additional role of the upstream oligo-C tract in promoting *URA8* transcription during CTP shortage. Notably, *URA2* and *URA8* induction is not always accompanied by a reduction in the abundance of their upstream CUTs, which led to the postulation of additional hypothetical regulatory models involving *ex novo* polymerase entry [44]. In summary, our data show that *URA8* regulation in response to inhibition of glutamine metabolism is highly specific to CTP, in analogy to GTP sensing [43].

## Involvement of transcription in DON-induced DNA breakage

The suppression of DNA breakage in *sml1Δmec1Δ* mutants by deletion of transcription regulators suggests that the DNA damage response kinase Mec1 coordinates DNA replication and transcription during DON treatment to avert DNA breakage. We hypothesize that DNA breakage could be prevented by two possible mechanisms: 1) Impaired transcription may slow down cell cycle and hence S-phase progression, potentially reducing dNTP consumption rate and providing more time to deal with shortage of NTPs that effect both RNA and DNA synthesis. 2) Modulation of transcription can reduce the formation of DNA-RNA hybrids [54] which represent replication obstacles that are harmful to DDR mutants [55,56]. DNA cleavage by nuclease could further enhance genotoxicity as such sites [57]. It will be interesting to further investigate if during DON treatment there is an accumulation of DNA-RNA hybrids or formation of topological constrains that cause the genome instability of DDR mutants.

## CTP synthase inducibility in human

The human genome contains two genes encoding CTP synthase enzymes (*CTPS1*, *CTPS2*), which are regulated through the formation of cytoophidia [51]. While a similar separation of roles into physiological and stress-induced CTP synthase is not known for these enzymes, a specific induction of CTPS1 but not CTPS2 after lymphocyte stimulation is required for B and T cell expansion [58], implying separate regulatory mechanisms. Notably, inactivating *CTPS1* mutations in human patients cause immunodeficiency [58], suggesting that rapid proliferation in this cell compartment is the physiological context of CTP depletion that requires adjustment of CTP synthase levels. Notably, the proliferation of *CTPS1*-deficient lymphocytes can be rescued by supply of extracellular cytidine *ex vivo*, suggesting that CTP synthesis through salvage can substitute for *de novo* synthesis but cannot provide sufficient CTP for rapid cell proliferation *in vivo*.

## Relevant contexts of glutamine metabolism inhibition

We describe chemogenomic interactions with the inhibition of glutamine metabolism, which is exemplified by the drugs DON and MSX. We hypothesize that similar interactions may be relevant in other settings of inhibition of glutamine metabolism. 1) Our results may be most predictive for treatment with the DON prodrug JHU-083 [6], which has the same intracellular mode of action as DON. 2) Combination treatments with PEGylated glutaminases efficiently lower the serum glutamine concentration and hence reduce glutamine availability for cancer cells [7]. 3) Asparaginases, which are routinely used in the treatment of Acute Lymphocytic Leukemia (ALL), hydrolyze both asparagine and glutamine in the serum [59]. 4) In addition,

the inhibition of c-Myc-regulated transporter SLC1A5 with the glutamine analog γ-glutamyl-p-nitroanilide reduces cancer cell proliferation *in vitro* and is a paradigm for glutamine starvation through reduction of its uptake [60,61]. 5) Novel drugs for the degradation of the epigenetic reader protein BRD4 offer an opportunity to deplete c-Myc in various cancers and dampen glutamine metabolism [62]. These treatments are similar to DON in reducing all intracellular glutamine-dependent reactions [2]. Importantly, our data show that inhibiting intracellular glutamine synthesis through MSX induces a CTP synthase response similar to DON, suggesting that the key findings of this study likely apply to targeting of glutamine metabolism in general. In this context, the synthetic hyper-sensitivity of *rtg1Δura8Δ* mutants to DON suggests that specific CTP synthase inhibition could strongly potentiate the effect of treatments aiming at intracellular glutamine depletion.

## Harnessing negative genetic interactions of CTP synthase

The multilayered glutamine analog responses (Fig 8) open up interesting possibilities of targeting several layers to enhance sensitivities synergistically. Specifically, the combination of Mec1$^{ATR}$ and CTP synthase inhibition imposes hyper-sensitivity to low DON concentrations. Mechanistically, CTP synthase inhibition renders cells dependent on Mec1$^{ATR}$ to handle replication stress and prevent chromosome breakage. While specific CTP synthase inhibition is not yet a current strategy in cancer therapy, inhibitors of nucleotide *de novo* synthesis (pemetrexed, methotrexate) reduce the levels of all nucleotides, including CTP [63]. The active triphosphate metabolite of the approved cancer drug gemcitabine, which is incorporated into DNA, also acts as potent inhibitor of CTP synthase [64]. In addition, approved immunosuppressant pyrimidine de novo synthesis inhibitors (leflunomide, mizoribine) reduce pyrimidine levels in the context of rapid proliferation and could facilitate CTP depletion, and repurposing for cancer treatment is under current investigation [65]. ATR inhibition is currently tested in various clinical trials [66]. Based on our observations, it would be very interesting to try the enhancement of glutamine starvation approaches with inhibition of ATR and CTP synthesis. We predict that pyrimidine nucleoside salvage and nucleobase uptake pathways may potentially add additional layers of complexity to treatment approaches targeting CTP synthase, which should be evaluated directly in cancer cells in the future. Notably, the inhibition of glutamine metabolism also impairs DNA integrity independently from nucleotide pools, by suppressing the production of glutamine-derived alpha-ketoglutarate, thereby suppressing DNA alkylation-reversing enzymes and boosting DNA alkylation levels, in particular in combination with alkylating chemotherapeutic agents [67]. We predict that this mechanism could contribute to the DON sensitivity of DDR mutants.

## Interactions between inhibition of glutamine metabolism and mTORC1

Our results highlight that inactivating mutations in TORC1 and glutamine transporters reduce the efficiency of the glutamine analog DON in Mec1$^{ATR}$ mutants. Both types of mutations partially reverse DON hyper-sensitivity and the activation of CTP synthase and DDR pathways. A reduction in the intracellular DON-to-glutamine ratio likely accounts for this observation. Assuming that DON and glutamine share similar plasma membrane transporters, which is supported by their structural similarity, transporter disruption would prevent DON uptake whereas glutamine is still synthesized intracellularly from glutamate. Inhibition of TORC1 is expected to de-repress the DON resistance transcription factor Rtg1 [29] and thereby enhance intracellular glutamine synthesis. Inhibitors of mTORC1 such as everolimus are promising reagents in cancers therapy [68]. Notably, cMYC upregulation has been reported repeatedly as adaptive resistance mechanism upon everolimus exposure [69–71]. Sphingoid base analog

FTY720, which acts as an anticancer agent in animal models, down-regulates mTORC1 activity and amino acid permeases in yeast and mammalian cells [72]. Hence, mTORC1 inhibition in a clinical context could enhance cMYC activity and suppress amino acid transporters. Our data suggest that this may have an impact on glutamine analog efficacy. In the light of new DON prodrugs which may enter clinical testing, it will therefore be of interest to investigate the interaction with these mTORC1 inhibitors.

## Materials and methods

### Strains and growth conditions

All strains used in this study are listed in S5 Table. All experiments were performed at 25˚C. Unless otherwise stated, yeast strains were grown in yeast extract/peptone with 2% glucose (YPD). Synthetic minimal medium for MSX treatments and nucleotide base restrictions contained 2% glucose, 1x yeast nitrogen base without amino acids and ammonium sulfate, 0.1% monosodium glutamate, 0.015% histidine, 0.015% tryptophan, 0.02% leucine, 0.004% uracil and 0.004% adenine.

### Cell treatments

For G1 synchronization, cells were incubated with 4 μg/mL α-factor (Genscript) for 150 min. For drug sensitivity assay, cells were grown to stationary phase, serial 1:6 dilutions were made, and one drop of each dilution was pin-spotted onto agar plates containing the indicated drugs. Plates were incubated for 2–3 days at 25˚C. Drugs were used at the following concentrations: 6-Diazo-5-oxo-L-norleucine (Merck Cat# D2141): as indicated, Hydroxyurea (Merck Cat# H8627): as indicated. Methionine sulfoximine (Merck Cat# M5379): 15 μM.

### Chemogenomic screens

Yeast mutant libraries for chemogenomic screens were prepared as described previously [73]. In brief, *sml1Δ*, *sml1Δmec1Δ* and *ura8Δ* query strains were constructed by gene targeting. The *sml1Δ* (2 replicates) and *sml1Δmec1Δ* (4 replicates) query strains were then crossed into the Yeast Knockout (YKO) library with deletion of non-essential genes [26], followed by diploid selection, sporulation, and selection for MATa and mutant alleles of query and library strains. The wt (4 replicates) and *ura8Δ* (4 replicates) query strains were crossed into a sublibrary composed of candidates from the screens conducted with *sml1Δ* and *sml1Δmec1Δ* queries. The selected strains were pin-spotted onto YPD agar plates containing the indicated drugs using ROTOR HDA (Singer Instruments). The concentration of DON was adjusted by screening aim and genotype (200 μM for the identification of resistance genes, 250 μM and 300 μM for identification of hypersensitivity suppressors in *sml1Δmec1Δ* and *ura8Δ* backgrounds, respectively). HU was used at 150 mM. Plates were scanned after 19 h– 42 h at 25˚C when colony sizes between treatments were comparable. Colony sizes were then quantified with the R package gitter [74]. Colony sizes were normalized by the intra-plate 70-percentile. Further position normalizations (column, row, neighborhood, competition) were performed as in [75]. We frequently observed spontaneous very large colonies in the *sml1Δmec1Δ* background on YPD + HU, which never occurred in all replicates and therefore represent spontaneous suppressors. We excluded these colonies from the analysis of all treatments in the *sml1Δmec1Δ* background using a maximum size exclusion filter on HU. Genetic interactors were called statistically significant if the distance between mean values was greater than two times the sum of standard deviations. To exclude candidates with very small effect sizes, we also applied a minimum filter for the standard deviation-corrected distance of mean values (0.05 for DON sensitivity and

resistance in *sml1Δ*, 0.5 for suppression of DON hypersensitivity in *sml1Δmec1Δ*). Based on functional similarity, we included a near-significant candidate (*hpt1Δ*) from the DON sensitivity screen with the *sml1Δ* query in the validation screen. We included an additional significance filter (Student's t-test with Benjamini-Hochberg correction, adjusted p-value < 0.1) for the validation screen of candidates obtained with the *sml1Δ* query, and of hypersensitivity suppressor screens with the *sml1Δmec1Δ* and *ura8Δ* queries. RStudio was used for data visualization. STRING (11.0) and Cytoscape (3.5.1) were used for network clustering and visualization. For plotting, the difference between experimental and reference values was corrected for the standard deviation as follows:

$$\frac{\Delta MEAN}{\left(1 + \left(2 \times \frac{\Sigma SD}{|\Delta MEAN|}\right)\right)}$$

where *ΔMEAN* is the difference between experimental and reference mean colony sizes and $\Sigma SD$ is the sum of standard deviations of experimental and reference colony sizes.

## Western blot analysis

Cells were fixed with 20% trichloroacetic acid (TCA) and disrupted by bead beating. Lysate and precipitate were mixed with 600 μL 10% TCA and pelleted. The pellet was resuspended in 1x Laemmli buffer with 5% β-mercaptoethanol and 160 mM Tris HCl (pH 7.4), boiled for 10 min and sonicated briefly. The extract was subjected to SDS gel analysis. The following antibodies were used: mouse monoclonal anti-Rad53 (clone EL7, in house [76], 1:5), mouse monoclonal anti-Pgk1 (Novex, Cat# 459250, 1:10.000), mouse monoclonal anti-HA (Biolegend, Cat# 901501, 1:10.000), rabbit polyclonal anti-Rnr3 (Agrisera. Cat# AS09574, 1:200), rabbit polyclonal anti-GST (in-house, 0.5 μg/ml), goat anti-mouse IgG (H + L)-HRP Conjugate (Bio-Rad, Cat# 1706516, 1:20000), goat anti-rabbit IgG (H + L)-HRP Conjugate (Bio-Rad, Cat# 1706515, 1:20000). Detection was performed by electrochemiluminescence (ECL, GE-Healthcare).

## Quantitative PCR analysis

Total mRNAs were extracted from $2x10^7$ cells using the RNeasy mini kit (QIAGEN). Reverse transcription was performed using the SuperScript VILO cDNA synthesis kit (Invitrogen) with 1 mg of total RNA. 1/80 of the cDNA reaction was used for quantitative PCR. Quantitative PCR was prepared using QuantiFast SYBR green PCR kit (QIAGEN) and run on the LightCycler 96 (Roche Life Science) according to the manufacturer's instructions. Relative cDNA quantification was performed. *TFC1* was used as normalization target as established previously [38]. Primers used for quantitative PCR are listed in S6 Table.

## Nano-LC-MS/MS metabolomics analysis

Metabolite extraction was performed as described previously [77]. In brief $10^8$ cells were collected by vacuum filtration using a 0.45μm pore size nylon membrane. The filter was positioned with cells side down in a petri dish containing 1 ml of extraction solution (LC-MS grade 40% acetonitrile, 40% methanol and 20% water, with 10 μM valine d8 (Cambridge Isotope Laboratories) as internal standard) cooled to -20˚C. The extraction was allowed to proceed for 15 min at -20˚C. Further extraction steps were performed on ice. The filter was washed with the collected extraction solution (10 times) and with 500 μl of fresh extraction solution. The extract was collected in a 1.5 ml Eppendorf tube, and the remaining material in the petri dish was collected with 250 μl of extraction solution. The pooled extract was

centrifuged for 5 min at 15.000 rpm in a microcentrifuge at -10˚C. The supernatant was transferred to a new 1.5 ml Eppendorf tube. The pellet was resuspended in 100 μl of extraction solution and kept on ice for 15 min. The sample was spin down and the supernatant was pooled with the other fraction.

Extracted samples were then diluted in either solvent A (acetonitrile) or extraction solution according to the applied chromatographic method. Nucleotide analysis was performed using a quadrupole Orbitrap QExactive-HF mass spectrometer (Thermo Fisher Scientific) coupled with a nanoLC Easy1000 system (Thermo Fisher Scientific). Chromatographic separation was achieved on a 75 μm i.d. fused-silica column (New Objective, Inc. Woburn, MA, USA), packed in-house with zwitterionic, polymer-based ZIC-pHILIC resin (5 μm, Sequant, kindly provided by Prof. Robert L. Hettich, ORNL Institute) [78] and zwitterionic Atlantis Premier BEH Z-HILIC resin (1.7 μm, Waters) using a high-pressure bomb loader (Proxeon, Odense, Denmark). The mobile phase was composed of acetonitrile (buffer A) and 10 mM triethylammonium bicarbonate (Sigma-Aldrich) and ammonium hydroxide (Sigma-Aldrich) in water (pH 9.6) (buffer B). The flow rate was set at 0.4 μL/min for the ZIC-pHILIC and 0.2 μL/min for the BEH Z-HILIC. The gradient for the ZIC-pHILIC column was the following: 80% A for 2 min, linear decrease to 20% of A in 10 min and 20% of A for 1 min (both solvents were then brought back to initial conditions in 1 min and maintained for 4 min). The gradient used for the BEH Z-HILIC was: 95% A for 2 min, linear decrease to 5% A in 15 min and 5% of A for 1 min (both solvents were then brought back to initial conditions in 1 min and maintained for 3 min). To avoid bias due to machine drift, samples were randomized and processed blindly. The mass spectrometer was operated in negative ion mode, full MS spectra were acquired in a mass range of 75–1000 m/z and a data-dependent top 5 MS/MS method was applied for metabolites identification and annotation. Ion source parameters were set as follow: voltage 2.5 kV, capillary temperature 300˚C and S-lens RF level of 50˚C. Full MS analysis was operated at 60000 resolution, $1e10^6$ of AGC target, 120 ms of maximum injection time. Data-dependent MS/MS was operated at 15000 resolution, $1e10^5$ of AGC target, 50 ms of maximum injection time, isolation window 2 m/z and normalized collision energies of 20, 50 and 100. Calibration curves of CTP, UTP, ATP and GTP (Sigma-Aldrich) were run to assess the quantification linearity range and as reference of annotation and retention time. Standards were prepared in buffer A and concentrations from 10 nM to 5 μM have been injected. XCalibur Qual Browser and XCalibur Quan Browser software (Thermo Fisher Scientific) were used to process and analyse the spectra.

## Flow cytometry analysis

For DNA content analysis, $1x10^7$ cells were pelleted by centrifugation and fixed with 70% ethanol, 250 mM Tris-HCl solution at pH 7.5. Cells were pelleted and treated with 1 mg/ml of RNAase A in 50 mM Tris-HCl solution at pH 7.5 and incubated overnight at 37˚C. After centrifugation, cells were resuspended in a staining solution (200 mM Tris-HCl pH 7.5, 200 mM NaCl, 80 mM MgCl$_2$, 0.05 mg/ml propidium iodide). Samples were diluted 1:10 in 1x PBS and analyzed with FACSCalibur (Becton Dickinson) and CellQuest software.

For the construction of reporter strains for flow cytometry, the endogenous *URA8* and *RNR3* genes in the library screening query strain were sequentially fused with a 2xDsRed and a GFP tag, respectively. The reporter strain was then crossed with the library mutants as described above to obtain mutant reporter strains. Cells were treated with drugs as described and fixed for 45 min (2% formaline, 50 mM Tris-HCl pH 7.4), washed twice (50 mM Tris-HCl pH 7.4) and stored in washing buffer at 4˚C in the dark until analysis. Cells were analyzed with Attune Nxt (Thermo Fisher), and data were visualized with FlowJo and RStudio software.

## Tetrad dissection analysis

Diploids were sporulated for 3d - 5d according to a standard protocol on VB agar (100 mM sodium acetate, 20 mM sodium chloride, 25 mM potassium chloride, 3 mM magnesium sulfate, 1.5% agar) at 23˚C. Tetrads were dissected using an MSM 400 dissection microscope (Singer) on YPD dishes. Haploids were cultured for 3d at 25˚C and colony sizes were quantified using ImageJ software.

## Pulsed field gel electrophoresis

Yeast cells treated with 0.1% sodium azide on ice, centrifuged and washed once with 50 mM EDTA pH 8.0. The pellet was resuspended in 50 μL SCE solution (1 M Sorbitol, 0.1 M Sodium Citrate, 0.06 M EDTA pH 8.0) per plug, and mixed with an equal volume of 50˚C molten Pulse Field Certified Agarose (BIO-RAD #162–0137). A plug-cast (BIO-RAD) was filled with 90 μL of cell/agarose mix per plug, and left for 20 min at RT and 10 min at 4˚C. For spheroplast preparation, agarose plugs were collected in a 50 mL polypropylene tube, covered with SCE solution with 0.2% β-mercaptoethanol and 1 mg/ml Zymolyase, and incubated at 37˚C for 1 h. The plugs were washed with an abundant volume of 0.5 M EDTA pH 8.0, resuspended in 0.5 M EDTA pH 8.0 with 0.1% Sarkosyl and 1 mg/ml proteinase K (0.5 mL/plug), and incubated overnight at 37˚C. The plugs were then washed three times with an abundant volume of 1x TE pH 8.0, transferred to a new 50 mL polypropylene tube and washed again with 1x TE pH 8.0 for 2 hours on a rotating wheel. Before electrophoresis, the plugs were equilibrated for 1 h in 0.5x Tris-borate with EDTA (TBE) on a rotating wheel. Electrophoresis was performed for 24 h at 10˚C in 1% (w/v) agarose containing 0.5x TBE using a CHEF-DR III Pulsed Field Electrophoresis Systems. Prior to image acquisition, gels were stained with 0.3 μg/ml ethidium bromide for 30 min. Image acquisition and analysis was performed using ImageLab 5.2 (BioRad) and imageJ.

## Statistics and Reproducibility

The following tests were applied: 1-way ANOVA with post hoc Tukey HSD test for multiple comparisons between groups of normally distributed, parametric data. Student's t-test (two-sided, unpaired) with Benjamini-Hochberg correction for multiple comparisons between groups for pairwise comparisons between groups of normally distributed, parametric data. Kruskal-Wallis rank sum test, followed by Mann-Whitney test (two-sided) with Benjamini-Hochberg correction for multiple comparisons between groups of not normally distributed data.

Where representative images are shown, we observed similar results in a total of three experimental repeats of the same clones (Figs 1B, 1C, 1F, 1H, 2A, 3C, 3D, 3F, 3G, 5A and 5B) or in three independent clones (Figs 1E, 2B, 3E, 4B and 4C), or two experimental repeats of the same clones (Figs 3A, 3B and 4E).

## Software

We used the following software: ImageJ (1.51d), STRING (11.0), Cytoscape (3.5.1), FlowJo (10.0.7r2), and RStudio (1.0.153) with R (3.4.1)

## Supporting information

**S1 Fig. Screen for glutamine analog resistance genes.** (A) Schematic of the validation screen for DON resistance genes. The heatmaps represent the difference between DON-treated vs. untreated colony size. All values are $\log_2$-transformed, corrected for standard deviation and

normalized to the 70-percentile of control clones. Statistical information is provided in S1 Table. (B) $10^7$ cells / mL of the indicated genotype in the S288C background were serially diluted (1:6), spotted on YPD plates with or without 250 μM DON or 200 mM HU and grown for 2d (Untreated), 3d (DON) or 4d (HU). (C, D) Exponentially growing cells of the indicated genotypes in the W303 (C) or S288c (D) background were treated with 300 μM DON for 2h as indicated. Metabolites were extracted and glutamine levels were quantified by nano-LC-MS/MS. Values are ion intensities (peak areas) normalized by the mean of untreated wt samples. Significance was calculated by Student's t-test (two-sided) over 5 replicate cultures. (E) Exponentially growing cells of the indicated genotypes in the S288c background were cultured in YPD and treated with 300 μM DON for 2h. Samples were collected to quantify *IMD2* mRNAs by RT-qPCR using *TFC1* as reference (n = 3 independent replicate cultures). Representative spot assay images are shown. wt = wildtype.
(TIF)

**S2 Fig. Flow cytometry analysis of the DON response in Nrd1 complex mutants.** (A) Strains with doxycycline-repressible alleles of the Nrd1 complex (*nrd1-tet*, *nab3-tet*) from the Tet-Promoters Hughes library were crossed with a query strain expressing endogenous DsRed-tagged Ura8 and GFP-tagged Rnr3. Mutant offspring was cultured overnight in YPD with 50 μM doxycycline, treated with 300 μM DON in YPD with 50 μM doxycycline and fixed with formaldehyde after 6h. Expression of the tagged proteins was analyzed by flow cytometry (n = 3 independent replicate cultures). The plots are from the samples in Fig 5G and provide additional information on cell parameters.
(TIF)

**S3 Fig. Screen for suppressors of glutamine analog hypersensitivity.** (A, B) The viable haploid gene deletion yeast library was crossed with *sml1Δ*, *sml1Δmec1Δ*, wt and *ura8Δ* query strains. Mutant offspring was selected, replicated on YPD (Untreated), YPD + 200 μM DON or YPD + 150 mM HU, and colony sizes were quantified. Each dot represents a library mutant. The x and y axes in (A) represent the relative colony sizes in the *sml1Δmec1Δ* background on YPD + HU vs. YPD (x) or YPD + DON vs. YPD (y) after 70-percentile normalization to all other *sml1Δmec1Δ* colonies with the same treatment, respectively. In (B) only the DON sensitivity suppressors from (A) are shown. The x and y axes in (B) represent the relative colony sizes in the *ura8Δ* (x) and *sml1Δmec1Δ* (y) backgrounds on YPD + DON vs. YPD after 70-percentile normalization to all other *ura8Δ* and *sml1Δmec1Δ* colonies, respectively. All values are log$_2$-transformed and adjusted for standard deviation. Significant sensitivity or resistance to DON in the *sml1Δmec1Δ* (A) or *ura8Δ* (B) background (see Materials and Methods) is indicated in blue (sensitive) and red (resistant). Statistical information is provided in S3 and S4 Tables.
(TIF)

**S1 Table. Yeast knock-out library colony sizes on YPD and YPD + DON in the sml1Δ background.**
(XLSX)

**S2 Table. Validation of yeast knock-out library colony sizes on YPD and YPD + DON after crossing with a wt query.**
(XLSX)

**S3 Table. Yeast knock-out library colony sizes on YPD, YPD + DON and YPD + HU in the sml1Δmec1Δ background.**
(XLSX)

**S4 Table. Yeast knock-out library colony sizes on YPD and YPD + DON in the ura8Δ background.**
(XLSX)

**S5 Table. Yeast strains used in this study.**
(XLSX)

**S6 Table. Primers used in this study.**
(XLSX)

## Acknowledgments

We thank Marco Foiani for financial support and mentoring, members of the Foiani laboratory and Mattia Pavani for helpful discussions, the Cogentech flow cytometry unit for experimental support, and Claudio Vernieri for comments on the manuscript.

## Author Contributions

**Conceptualization:** Arta Ajazi, Christopher Bruhn.

**Data curation:** Christopher Bruhn.

**Formal analysis:** Arta Ajazi, Ramveer Choudhary, Laura Tronci, Christopher Bruhn.

**Funding acquisition:** Arta Ajazi, Christopher Bruhn.

**Investigation:** Arta Ajazi, Ramveer Choudhary, Laura Tronci, Angela Bachi, Christopher Bruhn.

**Methodology:** Arta Ajazi, Laura Tronci, Christopher Bruhn.

**Project administration:** Christopher Bruhn.

**Resources:** Arta Ajazi, Christopher Bruhn.

**Software:** Christopher Bruhn.

**Supervision:** Arta Ajazi, Christopher Bruhn.

**Validation:** Arta Ajazi, Christopher Bruhn.

**Visualization:** Arta Ajazi, Laura Tronci, Christopher Bruhn.

**Writing – original draft:** Arta Ajazi, Christopher Bruhn.

**Writing – review & editing:** Arta Ajazi, Ramveer Choudhary, Laura Tronci, Angela Bachi, Christopher Bruhn.

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
