## [Decision Letter · Decision Letter 0]

9 Dec 2021

Dear Dr Bruhn,

Thank you very much for submitting your Research Article entitled 'CTP sensing and Mec1ATR-Rad53CHK1/CHK2 mediate a two-layered response to inhibition of glutamine metabolism' to PLOS Genetics.

The manuscript was fully evaluated at the editorial level and by independent peer reviewers. The reviewers appreciated the attention to an important problem, but raised some substantial concerns about the current manuscript. Based on the reviews, we will not be able to accept this version of the manuscript, but we would be willing to review a much-revised version. We cannot, of course, promise publication at that time.

If you decide to revise the manuscript for further consideration at PLOS Genetics, please aim to resubmit within the next 60 days, unless it will take extra time to address the concerns of the reviewers, in which case we would appreciate an expected resubmission date by email to plosgenetics@plos.org.

[LINK]

We are sorry that we cannot be more positive about your manuscript at this stage. Please do not hesitate to contact us if you have any concerns or questions.

Yours sincerely,

Alan G Hinnebusch

Guest Editor

PLOS Genetics

Gregory P. Copenhaver

Editor-in-Chief

PLOS Genetics

All three reviewers and the guest editor believe that this paper is potentially a valuable contribution to the field that would be suitable for this journal. While Rev. 1 and 3 had only minor comments, Rev. 2 and the guest editor had more substantive criticisms, some of which would likely require additional experimentation to address. Rev. 2 felt strongly that the authors have failed to cite critical published literature on the regulation of IMD2 by alternative start site selection dictated by the intracellular GTP concentration, and on relevant alternative start sites at URA8; and suggests that they have misinterpreted their results in their proposed model for URA8 regulation. He/she also voiced skepticism that up-regulation of URA8 transcription alone would be sufficient to account for DON-resistance and suggested that the authors test whether the Ura8 enzyme is distinguishable from Ura7 in being resistant to inhibition by DON, as shown previously for IMD2 and resistance to MPA. The guest editor felt that it would be important to show that deleting HPT1 confers a strong reduction in AMP/GMP levels and/or a strong induction of IMD2 transcription in cells treated with DON, to confirm the presumed mechanism of DON-sensitivity on deletion of HPT1. He/she also felt that categorization of the suppressors of DON toxicity based on their effects on UTP8 and RNR3 reporter expression +/- DON was not as clear-cut as suggested by the authors, and certain suppressors that don’t fall into either group were not discussed at all, calling for a more conscientious description of these data. Thirdly, it was felt that the analysis of these suppressors was quite preliminary and that a few follow-up experiments were required to show that representative suppressors of the first group actually do boost CTP levels in the presence of DON, whereas representative suppressors of the second group reduce the formation of ds DNA breaks (if this is indeed the presumed mechanism, which was not explicitly stated but should have been.)

Reviewer's Responses to Questions

**Comments to the Authors:**

Reviewer #1: The manuscript entitled « CTP sensing and Mec1ATR-Rad53CHK1/CHK2 mediate a two-layered response to inhibition of glutamine metabolism” by Ajazi and co-workers reports a chemogenetic analysis of the toxicity of a glutamine analog 6-diazo-5-oxo-L-norleucine (DON) in Saccharomyces cerevisiae. The authors identified several genes affecting DON resistance when mutated, they performed epistasis studies and focused on one of them URA8 which encodes an isoform of CTP synthase. They show that the other isoform (URA7) is less important for DON resistance although it is the major isoform. They establish that URA8, although contributing the minority isoform under standard growth conditions, is strongly induced under CTP deprivation, as for example when cells are treated with DON, and thereby is critical for resistance to the glutamine analog. The authors decipher a CUT-dependent mechanism of transcriptional ‘antitermination’ based on a repetition of Cs in the URA8 upstream region and the Nrd1/Nab3 complex. Finally, they show that in a ura8 background, DON treatment is associated to a DNA damage response and they screened for suppressors of this phenotype and this way identified mutants in the TOR pathway. This work gives an overview on the glutamine deprivation response and its connection to pyrimidine metabolism and DNA damage. This is a nice and solid piece of work. The experiments are sound, the results are clear and support the conclusions.

Minor point:

The experiments were done on YPD containing uracil and adenine. It would be interesting to compare the effect of DON on the mutants (most specifically ura8 and hpt1) on media containing or not uracil and adenine.

Reviewer #2: The manuscript from Ajazi et al. describes genetic interactions with the glutamine analog DON in budding yeast. Their results are of practical interest since pro-drug derivatives of DON are currently being tested as cancer therapies in animal studies, and the physiologically relevant molecular targets of DON are diverse. The genetic screens employed were limited to deletions of non-essential genes, but nevertheless yielded interesting results. A focus of the study is the surprising finding that deletion of one of two CTP synthetase genes, URA8, confers sensitivity to DON while deletion of the other, URA7, does not. (Glutamine is the source of nitrogen for the conversion of UTP to CTP by CTP synthetase.) In investigating the basis of this difference, the authors found that Ura8 protein is induced by glutamine or CTP depletion but Ura7 is not, and that induction occurs at the level of Ura8 mRNA synthesis. They therefore conclude that Ura8 confers resistance to DON via its increased expression, overcoming covalent inactivation of the enzyme by DON. This mechanism is similar to the resistance to mycophenolic acid (MPA) conferred by Imd2, but not by Imd3 and Imd4, except that Imd2 is also intrinsically resistant to MPA in addition to being induced. The authors do not investigate, or even consider, if Ura8 is intrinsically resistant to inactivation by DON. I am skeptical that induction of Ura8 to a level only slightly higher than constitutive Ura7 expression would confer DON resistance by mass action alone. The author should acknowledge this issue and, if possible, address it experimentally.

The authors unfamiliarity with the IMD2 and Nrd1-Nab3-Sen1 (NNS) termination literature causes them to misinterpret their results in other ways as well. In the Discussion (pg 17, line 4) the authors state: “So far it was not known if limitation of any NTP triggers a global induction for all CUT-regulated nucleotide synthesis enzymes, or if each NTP could regulate its respective synthesis pathway specifically.” This statement is incorrect. The authors need to familiarize themselves with the literature on yeast IMD2 regulation from the Brow and Reines labs published in 2008, which indicates that IMD2 is regulated by alternative start site selection that is dependent on intracellular GTP concentration. The upstream CUT start sites are at GG dinucleotides or GGG trinucleotides and are recognized when intracellular GTP is replete. The downstream mRNA start site is an AA dinucleotide. Thiebaut et al. 2008 (ref. 42, Fig. 6) mapped the URA8 upstream CUT starts to six sites, five of which are C nucleotides and the sixth is a T followed by a C, while the URA8 mRNA start sites are purines. Pyrimidine start sites are unusual for Pol II, so this pattern is striking and suggests that URA8 may be regulated by the same mechanism as IMD2 but sensing CTP rather than GTP concentration. The oligoC stretch mutated by the authors (Figure 5D) encompasses two of the start sites identified by Thiebaut et al. 2008 and the loss of induction observed supports an IMD2-like mechanism that is different from the mechanism proposed by the authors (Figure 5E). The authors’ proposed mechanism does not make sense to me. How would “crowding” of transcribing Pol II on the DNA displace the NNS complex from the nascent transcripts? It is scanning Pol II, not transcribing Pol II, that is responsive to GTP concentration on the IMD2 gene and, likely, on the URA8 gene.

Furthermore, the effect of NNS mutants on URA8 mRNA synthesis has already been examined by the Brow lab (Chen et al. 2017, Fig. S5). This study revealed the stronger effect of decreased Nab3 function compared to that of Nrd1, as is also seen by the authors (Fig. 5G). In general, the manuscript should be rewritten to better connect the current results with the published data for both IMD2 and URA8.

The experiments on effects of DON on the DDR pathway generally add value to the manuscript, although I question the identification of Rtg1 as a homolog (or even analog) of Myc.

Additional comments:

1) Figure 1H and elsewhere: arrows meant to indicate decreased levels should be placed alongside the compound (here, CTP) rather than above or below it, to prevent confusion with a metabolic pathway.

2) Page 7, lines 9-12: The difference in sensitivity of ura7-delta and ura8-delta to MSX (Fig. 2B) is considerably less than for DON (Fig. 1E) and may simply suggest that Ura8 has a lower Km for glutamine.

3) Figure 2B: the “sml1-delta” notation in this panel seems spurious. If not, it should be explained.

4) The Figure 3A legend should read “with or without 300 uM DON”.

5) Page 12, lines 11 and 12: IMD2 mRNA synthesis is regulated by GTP, not UTP (see above).

6) Page 16, line 42: the NNS termination complex does not “interrupt the synthesis of an upstream CUT”. It creates a CUT in place of an mRNA.

Reviewer #3: CTP sensing… Ajazi et al NAR

This is a thorough study examining the toxicity of glutamine analog called DON. The general motivation for this study is the examination of why glutamate analogs like DON are so toxic to cells. DON is a useful cancer therapeutic tool. The authors take advantage of the genetic and molecular tools in budding yeast, which is also sensitive to DON. Glutamine analogs (e.g. DON), which compete with glutamine as competitive inhibitors are toxic to cells due in part to inhibition of CTP synthase (encoded by two genes, URA7 and URA8). There are many cool details of the regulation of URA8 and URA7 that keeps the reader on his/her/their toes. The first screen searches for increased sensitization to DON treatment, and five genes are identified and studied. The second screen, introduced and analyzed in full in the last figure, searches for DON resistance or sensitivity in a mec1 sml1 mutant (defective in DDR responses). The model in Figure 7 reassures the reader that they got the story and the model straight, incorporating some of the genes identified into a conceptually coherent model.

This study has a lot in it, and specific studies seem well done and convincing. I reviewed this paper for another journal, I made several suggestions which the authors seemed to have followed. Aspects of the study remain a worthy challenge to follow conceptually, because there is a lot here. The writing seems clearer in this version, for which the readers will be thankful. And the Discussion is clearer to more readers than the earlier version. The data are spectacular. This paper is likely to provide a plethora of insights into how metabolic studies can be done; the full implementation of genetic techniques possible in yeast are on full display.

Minor Comments.

1. Fig 1J is …interesting. Sort of like it.

2. Fig 5E. Love the model. Yet, if they are showing the DNA and what is being transcribed, should not the oligoC be changed to oligoG as the template RNA polymerase is reading and stalling when CTP is low?

**Have all data underlying the figures and results presented in the manuscript been provided?**

Reviewer #1: Yes

Reviewer #2: Yes

Reviewer #3: Yes

PLOS authors have the option to publish the peer review history of their article (what does this mean?). If published, this will include your full peer review and any attached files.

Reviewer #1: No

Reviewer #2: No

Reviewer #3: No

---

## [Editor Report · Decision Letter 1]

14 Feb 2022

Dear Dr Bruhn,

We are pleased to inform you that your manuscript entitled "CTP sensing and Mec1ATR-Rad53CHK1/CHK2 mediate a two-layered response to inhibition of glutamine metabolism" has been editorially accepted for publication in PLOS Genetics. Congratulations!

Yours sincerely,

Alan G Hinnebusch

Guest Editor

PLOS Genetics

Gregory P. Copenhaver

Editor-in-Chief

PLOS Genetics

Comments from the reviewers (if applicable):

**Data Deposition**

http://datadryad.org/submit?journalID=pgenetics&manu=PGENETICS-D-21-01434R1

**Press Queries**

---

## [Editor Report · Acceptance letter]

27 Feb 2022

PGENETICS-D-21-01434R1 

CTP sensing and Mec1^ATR^-Rad53^CHK1/CHK2^ mediate a two-layered response to inhibition of glutamine metabolism 

Dear Dr Bruhn, 

We are pleased to inform you that your manuscript entitled "CTP sensing and Mec1^ATR^-Rad53^CHK1/CHK2^ mediate a two-layered response to inhibition of glutamine metabolism" has been formally accepted for publication in PLOS Genetics! Your manuscript is now with our production department and you will be notified of the publication date in due course.

With kind regards,

Zsanett Szabo

PLOS Genetics

On behalf of:
